# Discovery of a lectin domain that regulates enzyme activity in mouse *N*-acetylglucosaminyltransferase-IVa (MGAT4A)

Masamichi Nagae [1,2✉], Tetsuya Hirata[3], Hiroaki Tateno [4], Sushil K. Mishra [5], Noriyoshi Manabe [6], Naoko Osada[3,7], Yuko Tokoro[3], Yoshiki Yamaguchi[6], Robert J. Doerksen [5], Toshiyuki Shimizu [8] & Yasuhiko Kizuka [3✉]

*N*-Glycosylation is a common post-translational modification, and the number of GlcNAc branches in *N*-glycans impacts glycoprotein functions. *N*-Acetylglucosaminyltransferase-IVa (GnT-IVa, also designated as MGAT4A) forms a β1-4 GlcNAc branch on the α1-3 mannose arm in *N*-glycans. Downregulation or loss of GnT-IVa causes diabetic phenotypes by dys-regulating glucose transporter-2 in pancreatic β-cells. Despite the physiological importance of GnT-IVa, its structure and catalytic mechanism are poorly understood. Here, we identify the lectin domain in mouse GnT-IVa's C-terminal region. The crystal structure of the lectin domain shows structural similarity to a bacterial GlcNAc-binding lectin. Comprehensive glycan binding assay using 157 glycans and solution NMR reveal that the GnT-IVa lectin domain selectively interacts with the product *N*-glycans having a β1-4 GlcNAc branch. Point mutation of the residue critical to sugar recognition impairs the enzymatic activity, suggesting that the lectin domain is a regulatory subunit for efficient catalytic reaction. Our findings provide insights into how branching structures of *N*-glycans are biosynthesized.

[1] Department of Molecular Immunology, Research Institute for Microbial Diseases, Osaka University, Suita, Japan. [2] Laboratory of Molecular Immunology, Immunology Frontier Research Center (IFReC), Osaka University, Suita, Japan. [3] Institute for Glyco-core Research (iGCORE), Gifu University, Gifu, Japan. [4] Cellular and Molecular Biotechnology Research Institute, National Institute of Advanced Industrial Science and Technology (AIST), Tsukuba, Ibaraki, Japan. [5] Glycoscience Center of Research Excellence, Department of BioMolecular Sciences, University of Mississippi, University, Mississippi, MS, USA. [6] Division of Structural Glycobiology, Institute of Molecular Biomembrane and Glycobiology, Tohoku Medical and Pharmaceutical University, Sendai, Miyagi, Japan. [7] Graduate School of Natural Science and Technology, Gifu University, Gifu, Japan. [8] Faculty of Pharmaceutical Sciences, The University of Tokyo, Tokyo, Japan. ✉email: mnagae@biken.osaka-u.ac.jp; kizuka@gifu-u.ac.jp

Glycosylation, one of the most ubiquitous post-translational modifications in mammals, regulates a large variety of protein functions[1]. Glycans on proteins are biosynthesized in the endoplasmic reticulum (ER) and Golgi apparatus by stepwise and competitive actions of various glycosyltransferases[2], giving rise to an enormous number of glycan structures on glycoproteins[3]. A change in glycan structure leads to activation or dysfunction of its carrier protein, which is critically involved in various physiological and pathological events. This is exemplified by the fact that knockout of a certain glycosyltransferase gene in mice resulted in the disappearance of a specific glycan, leading to development or improvement of the disease-like phenotype, such as cancer[4], chronic obstructive pulmonary disease[5], diabetes[6], and dementia[7]. Furthermore, aberrant expression of a certain glycan structure is often correlated with disease progression, especially in cancer[8,9], which is clinically used as a biomarker and considered as a potential therapeutic target.

*N*-Glycans are a common class of glycans[10]. Early steps of *N*-glycan biosynthesis occurring in the ER are highly conserved among all eukaryotes to produce the common *N*-glycan intermediates, and late steps in the Golgi apparatus bring about species-, tissue-, protein-, and disease-specific *N*-glycans. In vertebrates, one of the most striking structural features of *N*-glycans is the variable number of GlcNAc branches (Supplementary Fig. 1a). The biosynthesis of the GlcNAc branches is catalyzed by the specific *N*-acetylglucosaminyltransferases (GnTs), GnT-I to -V[11–13] (Supplementary Fig. 1b). GnT-I (*MGAT1*) and -II (*MGAT2*) are required for the stepwise conversion of the *N*-glycans from immature (oligomannose and hybrid types) to mature forms (complex type) (Supplementary Fig. 1a). In contrast, GnT-III (*MGAT3*), -IVs (*MGAT4A* and *MGAT4B*), and -V (*MGAT5*) can compete for the common acceptor substrate (Supplementary Fig. 1a, GnGnbi). In addition, prior sugar transfer by GnT-III completely blocks subsequent action by either GnT-IVs or -V[14,15]. Furthermore, these enzymes show distinct tissue-specific expression and target protein selectivity[6,7], giving rise to complex branching patterns of *N*-glycans in each tissue and glycoprotein.

GnT-IV, unlike other GnTs, is composed of four homologous family members: GnT-IVa (*MGAT4A*), GnT-IVb (*MGAT4B*), GnT-IVc (*MGAT4C*, also known as GnT-VI), and GnT-IVd (*MGAT4D*, also known as GnT-1IP)[16–18]. Biochemically, GnT-IVa and -IVb enzymes catalyze the transfer of GlcNAc to α1-3-linked mannose of the core structure of *N*-glycan via the β1-4 linkage, whereas glycosyltransferase activity of mammalian GnT-IVc and -IVd has not been confirmed[19,20]. Double knockout of *Mgat4a* and *Mgat4b* in mice resulted in complete loss of both GnT-IV activity and its product glycans in tissues[21], demonstrating that GnT-IVa and -IVb are responsible for biosynthesis of the β1-4 GlcNAc on α1-3 mannose in mammals.

In contrast to the ubiquitous expression of GnT-IVb enzyme[21], GnT-IVa expression is specific to the gastrointestinal tissues, especially the pancreas[6]. Previous studies using knockout and transgenic mice for *Mgat4a* revealed that GnT-IVa is highly related to type 2 diabetes[6,22]. *Mgat4a*-deficient mice spontaneously develop diabetic phenotypes, such as high body weight and blood glucose, and impaired insulin secretion, compared with wild-type mice[6]. These abnormalities were also shown to be caused by aberrantly enhanced endocytosis of glucose transporter 2 (GLUT2) in β-cells. GLUT2 is a key molecule for both glucose uptake and insulin secretion, and modification of GLUT2 *N*-glycans by GnT-IVa is required for the efficient interaction between GLUT2 and galectins at the cell surface, leading to prolonged cell surface residency and glucose-sensing function of GLUT2[6]. Moreover, consumption of a high-fat diet in mice caused both transcriptional downregulation of *Mgat4a* and

diabetic phenotypes[22], and overexpression of GnT-IVa in mice was found to rescue these defects[22]. Furthermore, the mRNA levels of human *MGAT4A* were also shown to be reduced in pancreatic beta cells from diabetes patients[22]. These findings indicate that the control of *Mgat4a* expression and functional glycosylation of GLUT2 by GnT-IVa play key roles in the development of diabetes, and GnT-IVa could be a drug target. Furthermore, *MGAT4A* mRNA was shown to be aberrantly expressed in various cancer cells[23,24] and to promote their invasiveness by modulating the functions of glycoproteins, including integrin β1[25,26]. Together, these findings suggest that the design of compounds that can modulate GnT-IVa activity could lead to therapeutics.

Despite its biological significance, the basis for the catalytic reaction mechanism of GnT-IVa has not been elucidated. GnT-IVa (Q812G0 in UniProt) is composed of 526 amino acids with a predicted mass of 60.6 kDa, but the 3D structure, domain organization, and catalytic reaction mechanism of GnT-IVa remain completely unclear. Moreover, as described above, the design of GnT-IVa modulators could lead to the development of drug candidates. As such, it would be highly beneficial to obtain structural information about GnT-IVa[13]. In this study, by combining bioinformatic, biochemical, NMR, molecular dynamics, and crystallographic analyses, we discovered that the GnT-IVa enzyme has a unique lectin domain at its C-terminus, which is critical for its enzymatic reaction. Our findings provide insights into how GlcNAc branches are formed in *N*-glycans.

## Results

**Identification of C-terminal lectin domain in mouse GnT-IVa.** GnT-IVa, which biosynthesizes β1-4 GlcNAc branch (Fig. 1a), belongs to the GT54 family in the CAZy database[27] and is supposed to have a GT-A fold consisting of a single Rossmann fold. Although the Rossmann folds of mammalian GnTs such as GnT-I, II, and POMGnT1 are generally composed of ~10 α-helices and ~10 β-strands[28], secondary structure prediction of GnT-IVa suggested that there is an additional β-strand-rich region at the C-terminus (Supplementary Fig. 2). Furthermore, the Phyre 2 server[29] predicted that the central part of mouse GnT-IVa shows low sequence identity (17%) to GnT-II (Fig. 1b), and that the C-terminal region shows weak sequence homology to the carbohydrate binding module (lectin domain) of bacterial protein NagH, even though the amino acid sequence identity is only 14% (Fig. 1b). The lectin domain of NagH belongs to the CBM32 family and interacts with GlcNAcβ1-2Man[30], which is a part of the *N*-glycan. The crystal structure of the lectin domain of NagH in complex with GlcNAcβ1-2Man demonstrated that the disaccharide unit is buried inside the hydrophobic groove composed of Y819, W836, and W935, and the OH3 and OH4 of GlcNAc form hydrogen bonds with aspartic acid, D877 (Fig. 1c). Structure-based sequence alignment showed that these residues are all conserved in GnT-IVa (Y394, W410, W513, and D445) (Fig. 1c, right, and Fig. 1d, in red). Given these findings, we hypothesized that the C-terminal region of GnT-IVa also possesses lectin activity and is involved in the recognition of GlcNAc-containing glycans.

To first examine whether the C-terminal domain is involved in the enzymatic function of GnT-IVa, we designed several deletion constructs lacking either a part of the catalytic domain or the entire C-terminal domain (Fig. 1e, #1-4), purified these proteins from the culture media of COS7 cells (Fig. 1f, left), and performed enzyme assays using a fluorescently labeled *N*-glycan substrate (Fig. 1f, right). We confirmed that the soluble enzyme lacking the 77 N-terminal residues (#2) maintained its enzymatic activity, consistent with a previous report[15], while we found that the

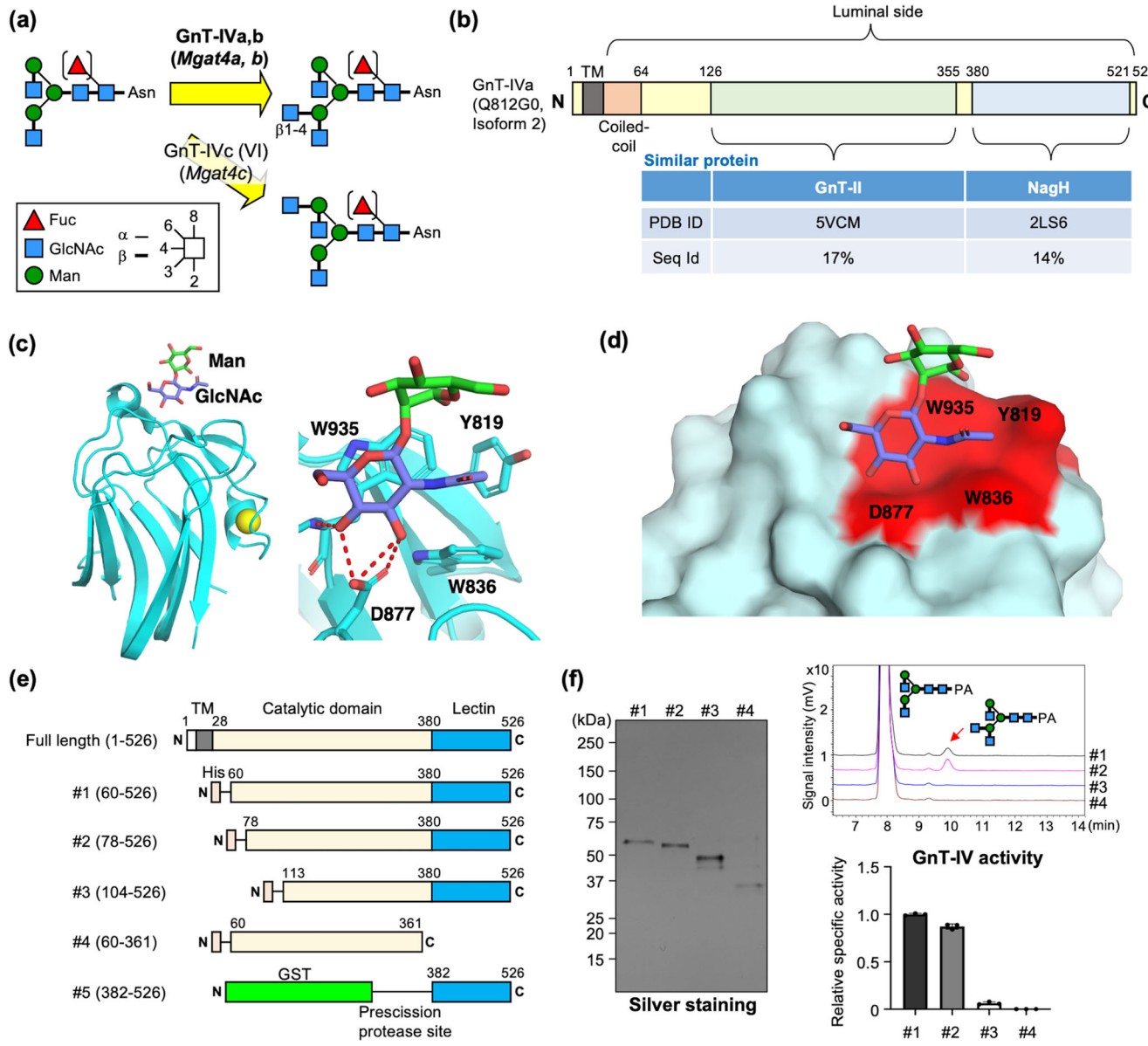

**Fig. 1 Presence of a lectin domain in the GnT-IVa C-terminal region. a** Schematic diagram of the GlcNAc transfer reactions catalyzed by GnT-IV family members. Activity of GnT-IVc has not been confirmed in mammals. **b** Proteins homologous to the central region and C-terminal region of mouse GnT-IVa. PDB ID and sequence identity to GnT-IVa are shown. **c** Overall structure of the carbohydrate binding module of NagH in complex with GlcNAcβ1-2Man (PDB code: 2WDB) (left) and close-up view of the sugar binding site (right). **d** The conserved amino acid residues in GnT-IVa are highlighted in red on the surface of NagH carbohydrate binding module in complex with GlcNAcβ1-2Man. **e** Constructs of mouse GnT-IVa used in this study. **f** Soluble GnT-IVa enzyme and its deletion mutants were purified from COS7 media. Purity of the proteins was checked by SDS-PAGE and subsequent silver staining (left). Enzyme activity of the proteins was measured by incubation with a PA-labeled acceptor sugar and analyzed by HPLC (right, upper). The specific activity of the proteins relative to that of #1 is shown ($n = 3$) (right, lower). The graph shows mean ± SD.

deletion of the entire C-terminal domain (#4) completely abolished the enzymatic activity of GnT-IVa (Fig. 1f, right), indicating that the C-terminal domain is necessary for its enzyme function.

Next, to test the above hypothesis that the C-terminal domain has lectin activity, we expressed the GST-tagged C-terminal domain (Fig. 1e, #5) in a bacterial system and purified it by affinity chromatography and gel filtration (Fig. 2a). Interestingly, the elution profile of the C-terminal domain in size exclusion chromatography with dextran sulfate polysaccharide resin showed large retardation compared with a molecular weight standard (Fig. 2a, right), and the retardation was largely canceled by the addition of GlcNAc but not glucose. This strongly suggests

that the C-terminal domain of GnT-IVa acts as a lectin with a preference for GlcNAc and strongly interacts with the poly-saccharide resin of the column.

**NMR and biochemical analyses of ligand preference of GnT-IVa lectin domain.** To confirm the direct sugar binding of GnT-IVa lectin domain, we next performed NMR titration experiments by collecting 2D $^1$H-$^{15}$N HSQC spectra of uniformly $^{15}$N-labeled GnT-IVa lectin domain with different protein-to-ligand ratios. A limited set of protein NH signals showed apparent chemical shift change by the addition of GlcNAc (Fig. 2b), reflecting the specific sugar binding to the lectin domain

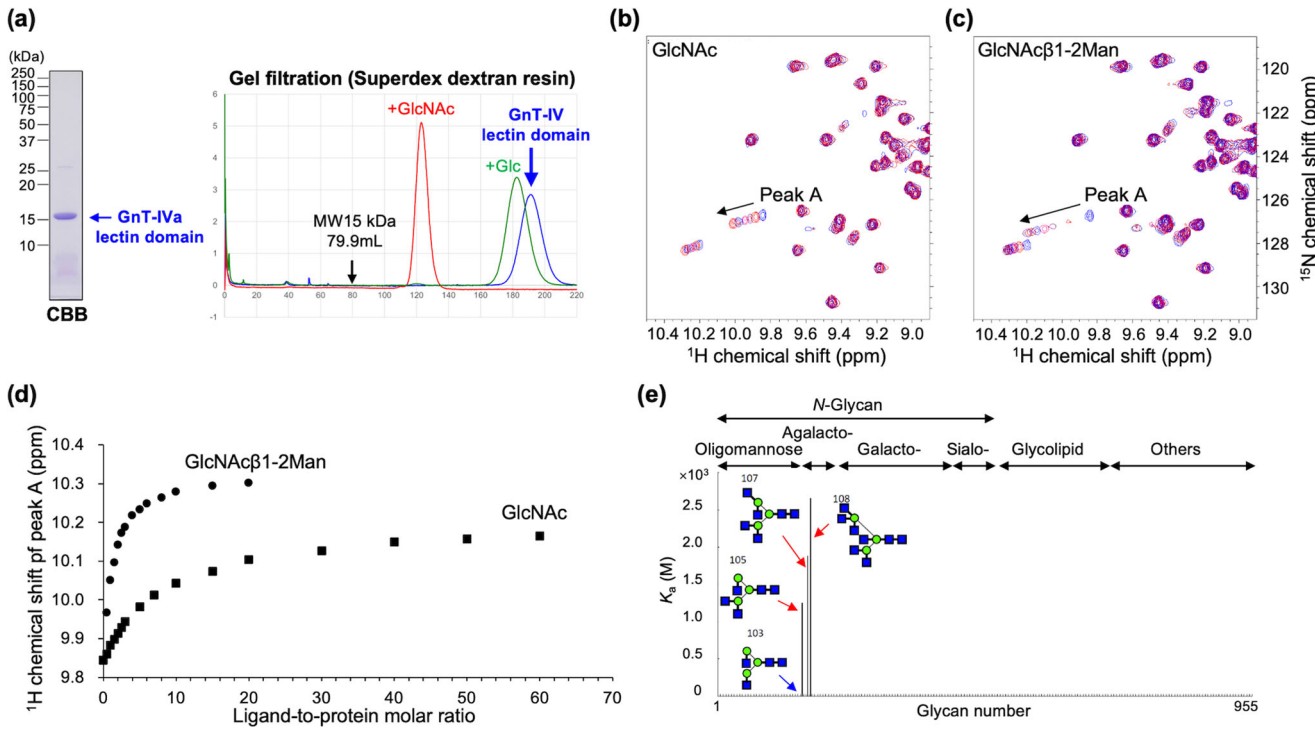

**Fig. 2 Glycan binding activity of GnT-IVa lectin domain. a** GST-GnT-IVa lectin domain (#5) was purified from *E. coli*, and GST tag was cleaved and eliminated. Purity was checked by SDS-PAGE and subsequent CBB staining (left). Elution profiles of the gel filtration analysis during the purification step are shown (right). The same gel filtration analysis was performed in the presence of 2 mM Glc (green) or GlcNAc (red). **b** Solution NMR analysis of the interaction between 15N-labeled GnT-IVa lectin domain and GlcNAc monosaccharide. Overlayed 1H-15N HSQC spectra of 0.4 mM [15N]GnT-IVa lectin domain at protein-to-ligand ratio of 1:0, 1:1, 1:2, 1:3, 1:5 and 1:7 for GlcNAc. Peak A was used for calculation of dissociation constant. **c** Solution NMR analysis of the interaction between 15N-labeled GnT-IVa lectin domain and GlcNAcβ1-2Man disaccharide. Overlayed 1H-15N HSQC spectra of 0.4 mM [15N]GnT-IVa lectin domain at protein-to-ligand ratio of 1:0, 1:0.5, 1:1, 1:1.5, 1:2, 1:2.5 and 1:3 for GlcNAcβ1-2Man. **d** Plot of 1H chemical shift (peak A) against ligand-to-protein molar ratio for GlcNAc (filled square) and GlcNAcβ1-2Man (filled circle). **e** Binding intensity of 157 glycans toward immobilized GnT-IVa lectin domain in frontal affinity chromatography.

without global conformational change. The binding process is in the fast exchange regime in terms of chemical shift and the dissociation constant was calculated as $3.1 \times 10^{-3}$ M using the NH peak A (1H 9.84 ppm, 15N 126.7 ppm) (Fig. 2b, d). The peak A was the one that was shifted the most by the addition of the disaccharide, GlcNAcβ1-2Man, in a dose-dependent manner (Fig. 2c), and the dissociation constant was calculated as $3.2 \times 10^{-4}$ M using peak A (Fig. 2c, d). This clearly indicates that the GnT-IVa lectin domain directly binds to GlcNAc-containing glycans and β1-2 linked Man largely contributes to its affinity.

To further evaluate the ligand preference in detail, we performed frontal affinity chromatography (FAC) analysis using immobilized GnT-IVa lectin domain and a panel of 157 glycans encompassing 88 *N*-glycans, 39 glycolipid-type glycans, and 30 other types of glycan (Supplementary Fig. 3). Surprisingly, GnT-IVa lectin domain showed highly specific binding with detectable affinities for only three glycans (#105 $K_a = 1.3 \times 10^3$ M$^{-1}$, #107 $K_a = 1.9 \times 10^3$ M$^{-1}$, and #108 $K_a = 2.7 \times 10^3$ M$^{-1}$), which commonly have β1-4-linked GlcNAc branches on the α1-3 branch, the product of GnT-IVa (Fig. 2e). In contrast, the affinity for an acceptor substrate biantennary glycan (#103) was below the detectable range (below $K_a$ ~$5 \times 10^2$ M$^{-1}$), even though it contains GlcNAcβ1-2Man (Fig. 2e). Furthermore, no other glycans tested showed detectable binding, including galactosylated forms of the enzyme product (#310 and #312). These findings indicate that the GnT-IVa lectin domain has the preference for short *N*-glycan products rather than *N*-glycan substrates or products that are further elongated.

**Crystal structure of GnT-IVa lectin domain displays high similarity to bacterial lectin**. To obtain the atomic details of the GnT-IVa lectin domain, we next aimed to solve its crystal structure. Because we could not obtain diffraction-quality crystals of the wild-type lectin domain despite repeated trials, we introduced an alanine mutation at D445, which corresponds to D877 in NagH and is supposed to directly interact with GlcNAc residue (Fig. 1c). The D445A mutant was successfully purified from bacterial cells (Fig. 3a) and showed no retardation in gel filtration analysis (Fig. 3b), indicating that the aspartate at position 445 of GnT-IVa also plays a critical role in the sugar binding, as observed in NagH. We successfully obtained well-diffracted crystals of D445A mutant and determined the crystal structure in unliganded form at 1.95 Å resolution (Table 1). The initial phases were determined by the iodide SAD method.

The overall structure of GnT-IVa lectin domain adopts a β-sandwich fold composed of nine β-strands with three short α-helices (Fig. 3c). Six molecules exist in the asymmetric unit of this crystal.

A DALI search revealed that the 3D structure of the GnT-IVa lectin domain shows high structural similarity to those of NagH [PDB code: 2W1U[30],] and IFT25/27 complex [PDB code: 2YC4[31],] (Fig. 3d). The structural superpositions with NagH (Z-score = 11.9) showed that the RMSD value of the corresponding 126 Cα atoms was 2.6 Å, whereas the superposition of IFT25, a component of the IFT25/27 complex (Z-score = 11.9), indicated that the RMSD of the corresponding 115 Cα atoms was 2.6 Å. As predicted from sequence analysis, the structural comparison with NagH GlcNAc-containing disaccharide (GlcNAcβ1-3GalNAc) complex demonstrated that

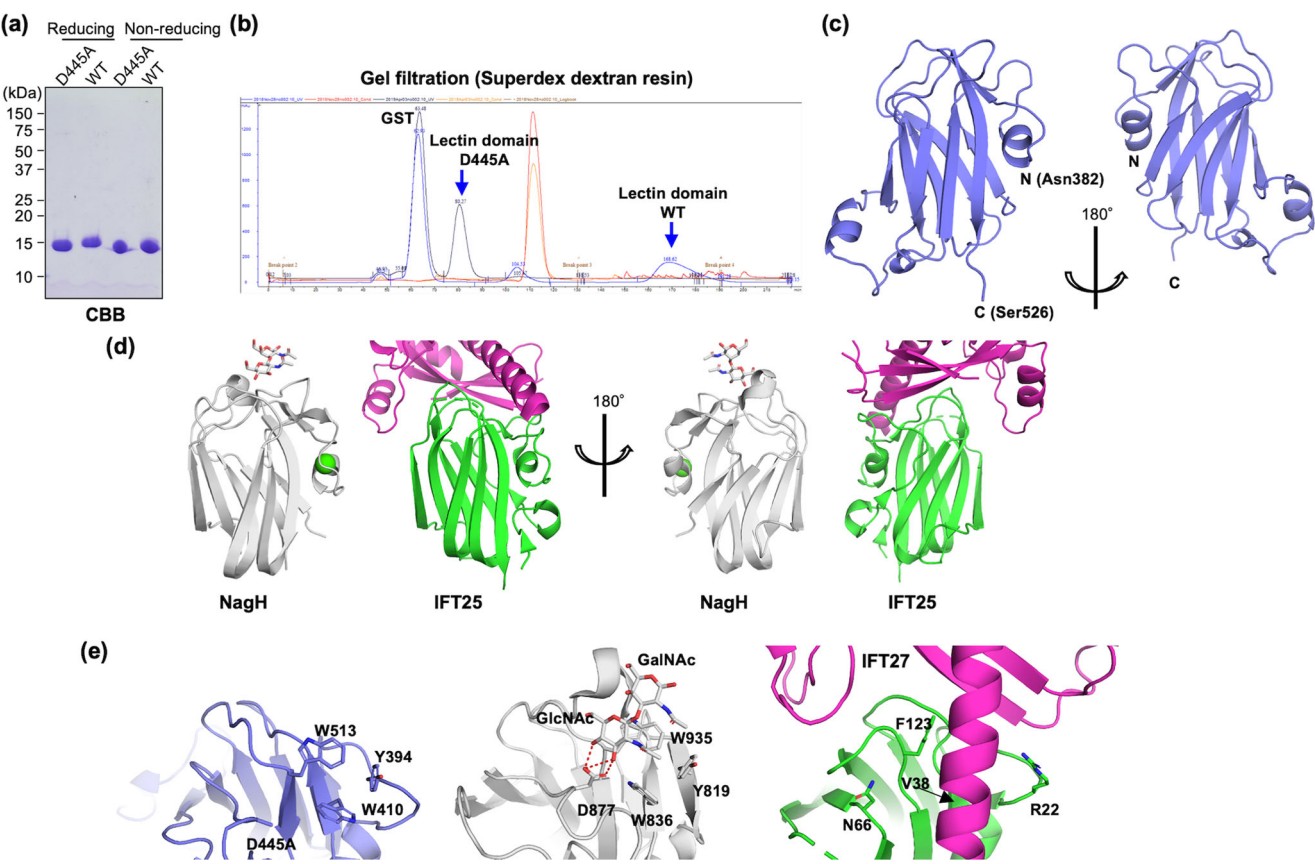

**Fig. 3 Structural analysis of GnT-IVa lectin domain. a** Purity of the GnT-IVa lectin domain and its D445A mutant prepared from *E. coli* was checked by SDS-PAGE and subsequent CBB staining. **b** Elution profiles of the gel filtration analysis during the purification step are shown. Protein absorbance ($A_{280}$) and ion conductance were colored in blue and red, respectively. The peak of ion conductance indicates the point of one column volume. **c** Overall structure of the lectin domain of mouse GnT-IVa D445A. **d** Structural neighbors of GnT-IVa lectin domain defined by DALI. Overall structures of NagH (PDB code: 2W1U[30], left panel) and IFT25 (PDB code: 2YC4[31], right panel), which are structurally similar proteins to the GnT-IVa lectin domain. These two proteins are viewed from the same angle as **c**. The carbohydrate and calcium ion are shown in stick and sphere models, respectively. **e** Close-up view of putative sugar binding sites of GnT-IVa (blue, left panel) and the corresponding regions of NagH CBM32 in complex with GlcNAcβ1-3GalNAc (PDB code: 2W1U, grey, middle panel) and IFT25 (PDB code: 2YC4, right panel). Four corresponding residues are shown in stick models and labeled.

amino acid residues that directly interact with the GlcNAc at the non-reducing end in NagH are completely conserved in the GnT-IVa lectin domain (Fig. 3e, left and middle panels). Four residues (Y819, W836, D877, and W935) are completely conserved in GnT-IVa (Y394, W410, D445, and W513). Thus, this indicates that the GnT-IVa lectin domain also binds to GlcNAc moieties in sugar ligands at the same site. However, several loop regions of GnT-IVa are slightly apart from those of NagH (Supplementary Fig. 4a). The position of D445A is slightly buried and there is no water molecule corresponding to the hydroxyl group of GlcNAc around D445A (Supplementary Fig. 4b). Note, though, that these loops contact neighboring molecules in the crystal packing, and such artificial interaction may affect the local structure (Supplementary Fig. 4c). In contrast, the overall structure of IFT25 is similar to that of the GnT-IVa lectin domain, but the corresponding region of the putative sugar binding site in IFT25 shows marked contrast with the other two proteins (Fig. 3e, right panel). The aspartate and three aromatic residues are not conserved. Instead, a long α-helix of IFT27 fully occupies the sugar binding site. This clearly explains the functional difference of GnT-IVa and IFT25.

**Molecular dynamics simulation suggests the contribution of product β1-4GlcNAc to the tight interaction with GnT-IVa lectin domain.** To understand the structural basis of glycan recognition by the GnT-IVa lectin domain, we performed molecular dynamics (MD) simulations and analyzed the potential for binding of a non-binding biantennary glycan (#103 in Supplementary Fig. 3, acceptor substrate) and three other glycans (#105, #107 and #108 in Supplementary Fig. 3, product glycans) which showed strong binding in FAC. To prepare for the calculations, we positioned each glycan to have its GlcNAc overlapped with the GlcNAc from the NagH X-ray crystal structure. Glycan #103 was able to be superpositioned well over the GlcNAc of the template, without any steric clashes with the protein atoms (Supplementary Fig. 5) but it drifted away from the binding site and became unbound during the subsequent MD simulations. By contrast, the other three glycans #105, #107 and #108 did show binding to the GnT-IVa lectin domain via their α1-3 arm in the MD simulations (Fig. 4a, b and Supplementary Fig. 6). In bisected glycan #108 the α1-6 arm is unlikely to bind to the lectin domain, as the α1-6 arm back-flips towards the chitobiose core and therefore is not exposed enough for recognition by proteins[32]. Assuming glycan binding to a particular lectin domain has a common molecular recognition mechanism for all the binding glycans, the likelihood of glycan binding to the lectin domain via the α1-6 arm is low and hence we omitted it. It is evident from MD that the α1-3 arm of #105, #107 and #108 can bind to the GnT-IVa lectin domain in a manner in which β1-2GlcNAc or β1-4GlcNAc occupies the primary GlcNAc binding site. The

molecular mechanics/generalized-Born surface area (MM/GBSA) binding energies show stronger binding affinity when β1-2GlcNAc occupies the primary binding site compared to β1-4GlcNAc (Fig. 4c; Supplementary Table 2). This suggests that the presence of β1-4GlcNAc in the α1-3 arm is needed for glycan binding, even if it may not be interacting directly with the lectin domain. It is likely that β1-4GlcNAc plays a key role in stabilizing the conformation of the overall glycan structure and contributes favorably to the entropic contribution. This explanation is supported by the glycan conformations in the protein bound states that show that the chitobiose core is more flexible and adopts two major conformations in the case of the non-binding glycan #103 compared to the case of the binders (Supplementary Fig. 7). This shows that the presence of β1-4GlcNAc in the α1-3 arm stabilizes the ligand conformation.

**Requirement of lectin domain in efficient enzyme reaction.** To investigate the functions of the GnT-IVa lectin domain in more detail, we purified soluble wild-type enzyme and its D445A mutant (Fig. 5a, b) and measured their enzymatic activity. The activity of D445A was reduced to ~62% of that of the wild type (Fig. 5c), indicating that the glycan binding ability of the lectin domain is required for the efficient enzymatic process. To further examine the role in the catalytic reaction, kinetic analyses were performed using various concentrations of the donor and acceptor substrates (Fig. 5d). D445A mutant showed a $K_m$ value toward UDP-GlcNAc comparable to that of the wild type, suggesting that the lectin domain is not involved in donor recognition. In contrast, the mutant showed a lower $K_m$ value toward the acceptor glycan (GnGnbi) than the wild type. This apparently higher affinity of D445A for the acceptor substrate might be derived from a slower catalytic reaction of the mutant.

We finally examined the significance of the lectin domain for protein glycosylation in cells. For this purpose, we established a GnT-IV-deficient cell clone by knocking out both *MGAT4A* and *MGAT4B* genes in HEK293 cells (Hirata et al., submitted elsewhere). The levels of GnT-IVa,b product glycans can be probed by *Datura stramonium* agglutinin (DSA) lectin, which showed reduced staining in the *MGAT4A/MGAT4B* double-knockout (DKO) cells compared with that in the wild type (Fig. 5e, 1st and 2nd lanes). The exogenous expression of full-length wild-type GnT-IVa in DKO cells recovered DSA staining, whereas that of D445A mutant caused weaker recovery of DSA signals for various glycoproteins (Fig. 5e). This demonstrates that the lectin domain of GnT-IVa is required for the efficient glycan biosynthesis toward glycoprotein substrates in cells.

## Discussion

In this study, we found that murine GnT-IVa has a lectin domain at the C-terminal region and shows unique ligand preference. Furthermore, it directly regulates the enzymatic activity and does not simply bind to glycans. The preference for the product glycan rather than the substrate raises the possibility that the lectin domain is involved in either prompt product release during the

### Table 1 Crystallographic data collection, phasing and refinement statistics.

|  | GnT-IVa lectin domain D445A mutant (PDB ID: 7VMT) | Iodide derivative |
| --- | --- | --- |
| **Data collection** |  |  |
| Space group | C2 | C2 |
| Cell dimensions |  |  |
| *a, b, c* (Å) | 82.9, 82.5, 148.9 | 82.7, 82.5, 149.4 |
| α, β, γ (°) | 90, 105.7, 90 | 90, 105.7, 90 |
| Resolution (Å) | 49.7-1.95 (2.06-1.95) * | 49.8-2.19 (2.26-2.19) |
| $R_{sym}$ | 7.0 (68.3) | 30.1 (490.4) |
| *I/σI* | 15.7 (3.0) | 27.1 (0.8) |
| Completeness (%) | 99.7 (98.9) | 98.6 (84.2) |
| Redundancy | 6.8 (6.7) | 88.3 (9.7) |
| **Refinement** |  |  |
| Resolution (Å) | 41.4-1.95 |  |
| No. reflections | 70,228 |  |
| $R_{work}/R_{free}$ | 20.8/24.4 |  |
| No. atoms |  |  |
| Protein | 6790 |  |
| Ligand/ion | 42 |  |
| Water | 210 |  |
| B-factors |  |  |
| Protein | 42.3 |  |
| Ligand/ion | 30.8 |  |
| Water | 37.2 |  |
| R.m.s deviations |  |  |
| Bond lengths (Å) | 0.008 |  |
| Bond angles (°) | 0.932 |  |

*Values in parentheses are for highest-resolution shell.

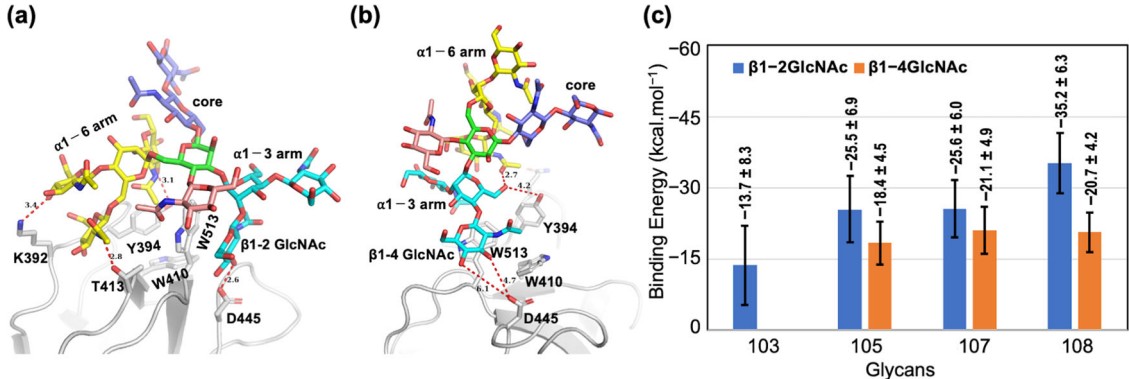

**Fig. 4 Calculated binding energies and binding modes of glycans with the GnT-IVa lectin domain. a, b** The binding mode of glycan #108 after 500 ns molecular dynamics in the **a** β1-2GlcNAc•••D445 or **b** β1-4GlcNAc•••D445 binding mode. The glycan carbons are colored differently for clarity: α1-3 arm (C cyan), α1-6 arm (C yellow), bisecting GlcNAc (C pink) and chitobiose core (C purple). Red dashed lines show polar interactions between glycan #108 and amino-acid residues within 6 Å. **c** MM/GBSA energies of acceptor substrate biantennary glycan (#103) and product glycans (#105, #107 and #108) in two different binding modes in which a different substituent of the α1-3 arm interacts with D445 in the binding site: (i) β1-2GlcNAc (blue bars) or (ii) β1-4GlcNAc (orange bars). The graph shows means ± standard deviations of the free energies (500 frames).

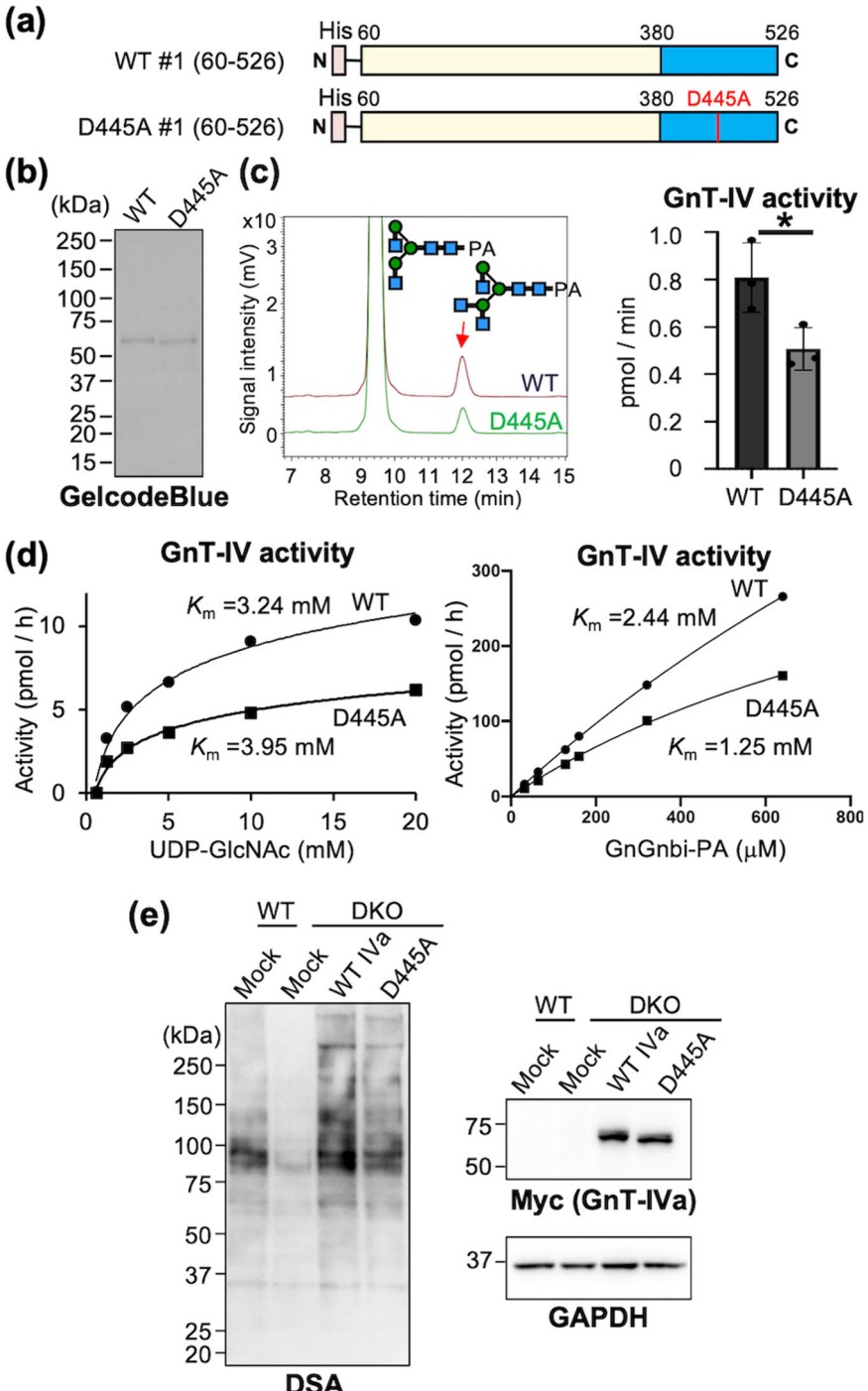

**Fig. 5 Requirement of the lectin domain for GnT-IVa-mediated glycosylation. a** Constructs used for soluble GnT-IVa enzyme and its D445A mutant. **b** Purity of the enzymes prepared from COS7 media was checked by SDS-PAGE and subsequent GelCode Blue staining. **c** Enzyme activity of the proteins was measured by incubation with the PA-labeled acceptor sugar and analyzed by HPLC (left). The specific activities of the proteins are shown ($n = 3$) (right). The graph shows mean ± SD (*$p < 0.05$, Mann–Whitney U-test). **d** Kinetic analyses of WT GnT-IVa and D445A mutant were performed using various concentrations of the donor substrate (left) and the acceptor substrate (right). **e** Proteins from mock-treated HEK293 wild-type and *MGAT4A/MGAT4B* double-knockout (DKO) HEK293 cells transfected with empty vector (mock), WT GnT-IVa-myc vector, or GnT-IVa D445A-myc vector were subjected to SDS-PAGE and blotted with HRP-conjugated DSA (left), anti-myc antibody (upper right), or anti-GAPDH antibody (lower right).

catalytic reaction cycle or the efficient modification of glycoproteins in which β1-4GlcNAc already exists (Fig. 6).

Previous studies revealed that GnT-IVa is involved in diabetes[6,22], suggesting that clarifying the regulation mechanisms of GnT-IVa activity could lead to development of a new strategy for the treatment of diabetes. Furthermore, one coding SNP (T236A) of human GnT-IVa (*MGAT4A*) was also reported in the ClinVar database[33]. We found that mutation of the corresponding residue in mouse GnT-IVa (T227A) resulted in a decrease in activity in cells and in vitro enzyme assays (Supplementary Fig. 8). Although the relevance of this SNP in specific disease is unclear at present, the reduction in activity by this coding SNP could possibly be involved in development or exacerbation of diseases.

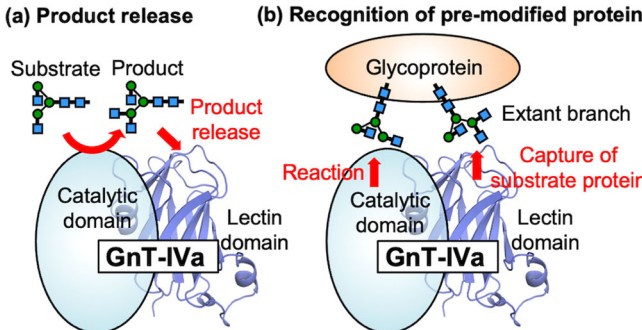

**Fig. 6 Schematic model of the role of GnT-IVa lectin domain.** Our data showed that the lectin domain specifically binds *N*-glycan products and is required for efficient GnT-IVa reaction. We propose that the lectin domain is involved in either (**a**) the prompt product release in the catalytic reaction cycle or (**b**) the efficient modification of glycoproteins in which β1-4GlcNAc already exists.

Several mammalian glycosyltransferases acting on other types of glycans contain lectin domains within their polypeptides. UDP-GalNAc:polypeptide *N*-acetylgalactosaminyltransferases (GALNTs) in the GT27 family have an N-terminal GT-A fold and a C-terminal lectin domain connected by a flexible linker[34–37]. The 3D structures of GALNT lectin domains are similar to those of ricin B-type lectins classified into the CBM13 family. The GALNT lectin domains adopt β-trefoil folds and contain three tandem repeats (α, β, and γ) that bind to GalNAc on product glycopeptides and critically guide catalysis to unmodified acceptor sites for efficient glycosylation[38,39]. In contrast, POMGnT1, another GlcNAc transferase acting on *O*-mannose, has both an N-terminal lectin domain and a C-terminal GT-A fold[40]. Although the lectin domain of POMGnT1 also shows binding affinity for the product, terminal β-linked GlcNAc, the lectin domain appears to be unrelated to the enzymatic activity and is suggested to function in the recruitment of this enzyme and its interacting partner to the substrate glycoprotein. Our results indicate that the GnT-IVa lectin domain shows ligand preference toward the product GlcNAcβ1-2[GlcNAcβ1-4]Man (Fig. 2e), and the D445A mutation of GnT-IVa dramatically impairs the catalytic activity toward an oligosaccharide (Fig. 5c, e). These findings suggest that the lectin domain of GnT-IVa positively modulates the catalytic cycle rather than just capturing substrate protein bearing an extant β1-4 GlcNAc branch (Fig. 6). Another possibility is that the binding of the product glycan to the lectin domain could induce a conformational change of the catalytic domain for efficient catalysis. To clarify these points, it is important to solve the entire structure of GnT-IVa in complex with the product glycan in future work.

Like GnT-IVa, several GlcNAc transferases such as GnT-I, II, III, and V are involved in *N*-glycan maturation[13]. Although all these enzymes are inverting glycosyltransferases that use UDP-GlcNAc as donor substrates, there is wide variety in the domain architecture. GnT-I has GT-A folds with an extended domain at the C-terminus[41], while GnT-II has inserted loops named Loop-Helix-Loop (LHL) in the middle of the GT-A fold[28]. GnT-III is predicted to have a GT-A fold with DXD motif and an inserted glycine/proline-rich region at the N-terminal region of the catalytic domain[13]. GnT-V is a unique GlcNAc transferase that has a GT-B fold with two accessory domains and shows completely different domain architecture[42]. The N-terminal accessory domain is involved in subcellular localization[43,44] and recognition of glycoprotein substrates[45], while the physiological function of the C-terminal domain remains unclear. The catalytic domain of GnT-IVa is also supposed to be a GT-A fold and similar to GnT-II

rather than GnT-I; thus, the catalytic domain of GnT-IVa directly connects to the C-terminal lectin domain via a slightly long linker (~25 residues) (Supplementary Fig. 9a). Because deletion of part of this linker completely abolished the enzymatic activity (Supplementary Fig. 9b, c), the distance between the catalytic domain and the lectin domain could be critical. GnT-IVa is missing the LHL insertion, which in GnT-II interacts with GlcNAcβ1-2Manα1-3Man, named as a "recognition arm," of the acceptor *N*-glycan, suggesting the different acceptor recognition mode of GnT-IVa compared with that of GnT-II. It is possible that GnT-IVa directly recognizes the recognition arm via its catalytic pocket.

There are four GnT-IV isoforms (GnT-IVa–d) in mammals. GnT-IVb shows the same branching activity in vitro as GnT-IVa, but with weaker affinity to both donor and acceptor substrates[15], and is rather ubiquitously expressed among organs. In mice with double deficiency of *Mgat4a* and *Mgat4b*, GnT-IV activity is completely abolished in all tissues, resulting in the disappearance of the GlcNAcβ1-4 branch on the α1-3 arm[21]. This demonstrates that only GnT-IVa and GnT-IVb work as active GnT-IV enzymes and that GnT-IVc (GnT-VI) and GnT-IVd do not contribute to the synthesis of the branch. The mouse GnT-IVb shares ~60% amino acid residues overall and 53% in the lectin domain with GnT-IVa. Of note, four amino acid residues (Y394, W410, D445, and W513 in IVa) at the sugar binding sites are completely conserved in these two enzymes (Supplementary Fig. 10), suggesting that the C-terminal domain of GnT-IVb also acts as a lectin domain. Human GnT-IVc encoded by *MGAT4C*, also known as GnT-VI or GnT-IV-H, was cloned from the commonly deleted region in pancreatic cancer at 12q21[46]. This gene was found to be highly expressed in adult brain. No enzyme activity of human GnT-IVc has yet been detected and the physiological function of this protein remains unclear. GnT-IVd encoded by the *MGAT4D* gene is also known as GnT1IP-L and inhibits the activity of GnT-I via its luminal domain[47]. Multiple sequence alignment showed that all four members share the putative catalytic residue (D313 in mouse GnT-IVa) and EDD motif, which is a putative donor binding site (Supplementary Fig. 10). In addition to the catalytic domain, GnT-IVa, IVb, and IVc have lectin domains, while GnT-IVd completely lacks one. GnT-IVc also has a lectin domain, but Y394 located at the putative sugar binding site is not conserved. This suggests that the GnT-IVc lectin domain may have lost the ability to bind sugar. Fish and chicken orthologs of human *MGAT4C* encode GnT-VI enzymes, which transfer GlcNAc to the OH4 position of the Manα1-6 arm of the core structure of *N*-glycan, forming the most highly branched penta-antennary glycans in these organisms[18,48]. In mammalian tissues, the presence of GnT-VI activity and its product glycans have yet to be definitively confirmed. The roles of the lectin domain in this enzyme family warrant future detailed investigation.

In this study, we discovered a lectin domain at the C-terminal region of GnT-IVa. This domain shows structural and functional similarities with bacterial CBM32 and is positively involved in the catalytic reaction cycle. Our findings suggest the cooperative catalytic reaction mechanism of GnT-IV mediated by the catalytic and lectin domains, providing insights into how *N*-glycans are branched in mammalian cells.

## Methods

**Materials**. Chemical compounds and crystallization reagents were purchased from Nacalai Tesque, Inc. and Hampton Research Corp, respectively. Anti-myc (05-724) and anti-GAPDH (MAB374) antibodies were purchased from Millipore. DSA lectin (J105; J-Chemical) was labeled with HRP using a peroxidase labeling kit – NH$_2$ (DOJINDO), following the manufacturer's procedure.

**Plasmid construction**. All primers used for DNA construction are listed in Supplementary Table 1. cDNA encoding full-length mouse GnT-IVa was amplified by PCR using C57BL/6 mouse pancreas cDNA library as a template, and the amplified

fragment was cloned into pCR-Blunt II-TOPO. For the expression of full-length mouse GnT-IVa, a DNA fragment was amplified and inserted into NotI/XhoI sites of pcDNA6/mycHisA. For enzymatic assays, cDNAs encoding the luminal domain of GnT-IVa (60–526, 78–526, 104–526, 60–361) were amplified by PCR and ligated into EcoRV/XhoI sites of pcDNA-IH[49] for the expression of N-terminally His-tagged proteins. cDNAs encoding the luminal domain of GnT-IVa lacking a part of the linker (Δ4, Δ9, and Δ19) were amplified by PCR and ligated into PstI/XhoI sites of pcDNA-IH. The plasmids for soluble (pcDNA-IH/GnT-IVa 60–526) and full-length (pcDNA6/mycHisA/GnT-IVa) D445A mutants and full-length T227A mutant of GnT-IVa were constructed using QuikChange Lightning Site-directed Mutagenesis Kit (Agilent), in accordance with the manufacturer's protocol. cDNA encoding the mouse GnT-IVa lectin domain (382–526) was incorporated into the pGEX6P-1 vector and expressed as a glutathione S-transferase (GST)-fused protein.

### Expression, purification, and crystallization of GnT-IVa lectin domain.

The GST-fused gene (pGEX6P-1/mGnT-IVa lectin domain) was transformed into the *Escherichia coli* BL21(DE3) RIPL strain. The transformed cells were cultivated at 37 °C until the $OD_{600}$ reached 0.5, after which the expression was induced by adding 0.5 mM IPTG. The cells were cultivated at 18 °C for 16 h.

Cells were suspended in lysis buffer [0.1 M Tris-HCl (pH 8.0), 0.5 M NaCl] and disrupted by sonication. The supernatant of the lysate was collected by centrifugation (48,000 × g, 20 min) and loaded onto glutathione Sepharose 4B (GS4B; GE Healthcare), followed by washing with the lysis buffer. The bound protein was eluted in the lysis buffer containing 25 mM reduced glutathione. The GST tag was cleaved by PreScission Protease digestion. The cleaved proteins were subjected to gel filtration chromatography using a Superdex75 column (GE Healthcare). The elution peak was collected and concentrated up to 5.6 mg/mL by Amicon Ultra (MWCO 3 K). The purities of proteins were checked by SDS-PAGE with Coomassie Brilliant Blue (CBB) staining (Fig. 3a).

All crystallization trials of the lectin domain D445A mutant (residues 382–526) were performed by the sitting drop vapor diffusion method at 20 °C. The crystals in unliganded form were obtained under the conditions of 0.1 M Bis-tris (Bis(2-hydroxyethyl)iminotris(hydroxymethyl)methane, pH 6.5), 0.2 M magnesium chloride, and 25% (w/v) polyethylene glycol 3350. These crystals were directly flash-cooled in liquid nitrogen. Iodide derivatives were generated by immersing the D445A mutant crystals in 0.1 M Bis-tris (pH 6.5), 0.2 M magnesium chloride, 25% (w/v) polyethylene glycol 3,350, and 0.1 M potassium iodide for a few minutes prior to flash cooling.

X-ray diffraction data sets were collected at the synchrotron radiation source at AR-NE3A in the Photon Factory (Tsukuba, Japan) and BL45XU in SPring-8 (Harima, Japan). All data sets were processed and scaled using the program XDS[50]. Initial phase determination was performed by the single anomalous diffraction (SAD) method using iodide derivatives with the program Crank2[51] of the CCP4 program suite. Model building was performed manually using the program COOT[52]. Refinement was initially conducted using REFMAC5[53] and Phenix.refine of the Phenix program suite with individual B-factor options[54] for the final model. One out of six molecules in the asymmetric unit has relatively high temperature factors due to the loose crystal contact, but we treated the molecule as full occupancy. The stereochemical quality of the final models was assessed by MolProbity[55]. Data collection, phase determination, and refinement statistics are summarized in Table 1. Structure factors and atomic coordinates of the GnT-IVa lectin domain have been deposited in Protein Data Bank under accession code 7VMT. All structures were depicted using PyMOL (The PyMOL Molecular Graphics System, Version 2.0, Schrödinger, LLC).

### NMR titration analysis using $^{15}$N-labeled GnT-IVa lectin domain.

Uniformly $^{15}$N-labeled GnT-IVa lectin domain was prepared by culturing transformed BL21(DE3) cells with SPECTRA 9 ($^{15}$N, 98%) medium (Cambridge Isotope Laboratories). $^{15}$N-labeled lectin domain was dissolved in 20 mM sodium phosphate buffer (pH 6.0) at a protein concentration of 0.4 mM in a volume of 0.6 ml. All NMR experiments were carried out on a Bruker 800 MHz spectrometer equipped with a triple resonance cryogenic probe. The probe temperature was set to 298 K. $^{1}$H chemical shifts are given in ppm calibrated with external reference DSS (4,4-dimethyl-4-silapentane-1-sulfonic acid) at 0 ppm, and $^{15}$N chemical shifts are referenced using chemical shift referencing ratio ($^{15}$N/$^{1}$H 0.101329118)[56]. Titration experiments were conducted using N-acetyl-D-glucosamine (GlcNAc) or GlcNAcβ1-2Man disaccharide. A series of 2D $^{1}$H-$^{15}$N HSQC spectra were collected at protein-to-ligand molar ratio of 1:0, 1:0.5, 1:1, 1:1.5, 1:2, 1:2.5, 1:3, 1:5, 1:7, 1:10, 1:15, 1:20, 1:30, 1:40, 1:50 and 1:60 for GlcNAc and 1:0, 1:0.5, 1:1, 1:1.5, 1:2, 1:2.5, 1:3, 1:4, 1:5, 1:6, 1:8, 1:10, 1:15 and 1:20 for GlcNAcβ1-2Man. Data processing and spectral display was performed using TopSpin Version 4 (Bruker). Dissociation constants for GlcNAc and GlcNAcβ1-2Man were calculated using the changes in $^{1}$H chemical shift of the protein NH signal ($^{1}$H 9.84 ppm, $^{15}$N 126.7 ppm) assuming that ligand binding is in the fast exchange regime.

### Frontal affinity chromatography (FAC).

FAC analysis was performed as described previously[57]. Briefly, the lectin domain in GnT-IVa was immobilized on NHS-activated Sepharose 4 Fast Flow (GE) at a concentration of 5 mg/ml and packed into a miniature column (inner diameter, 2 mm; length, 10 mm, bed volume, 31.4 μl; Shimadzu) and connected to an automated FAC system. A panel of

pyridylaminated (PA) and p-nitrophenol (pNP) glycans was successively injected into the columns by the auto-sampling system, and the elution was detected by fluorescence (excitation, 310 nm; emission, 380 nm) or absorbance at 220 nm. The elution front of each glycan relative to that of an appropriate control, referred to as $V-V_0$, was then determined. Analysis of the concentration dependence was performed using Core6-pNP to obtain the $Bt$ value.

### Enzymatic activity assay.

Purification of recombinant enzymes and activity assays were performed as described previously[42] with modifications. N-terminally His-tagged enzymes were expressed in COS7 cells and purified from culture media through a Ni$^{2+}$-column. The purified enzymes were mixed with 100 pmol PA-labeled GlcNAc-terminated biantennary glycan (GnGnbi-PA) in 10 μl of a buffer containing 125 mM MES, pH 6.25, 10 mM MnCl$_2$, 200 mM GlcNAc, 0.5% (v/v) Triton X-100, 1 mg/ml BSA, and 20 mM UDP-GlcNAc. For kinetic analysis with the donor substrate, the reaction was carried out in the presence of 10 μM acceptor substrate (GnGnbi-PA) and 20, 10, 5, 2.5, 1.25, or 0.625 mM donor substrate (UDP-GlcNAc) for 15 min at 37 °C. For the acceptor substrate, the reaction was carried out in the presence of 20 mM donor substrate and 640, 320, 160, 128, 64, or 32 μM acceptor substrate for 15 min at 37 °C. The enzyme reaction was stopped by boiling for 3 min and 40 μl of water was added to the mixture, followed by centrifugation at 15,000 × g for 5 min. The supernatant was analyzed by reverse-phase HPLC (Prominence; Shimadzu) equipped with an ODS column (TSKgel ODS-80TM; TOSOH Bioscience). Kinetic parameters were calculated using GraphPad Prism 8 (GraphPad Software).

### Computational Modeling.

The structure of the GnT-IVa inactive mutant (D445A) obtained from X-ray crystallography in this work was used as a starting point for calculations to help understand the molecular basis of glycan recognition by the wild type lectin domain. We performed A445D mutation in the GnT-IVa lectin domain using the protein mutation wizard in PyMOL to obtain a model of the structure of the wild-type. Glycans #103, #105, #107 and #108 were prepared using glycam-web[58] and the obtained minimum energy conformers were used to study binding to the lectin domain. The structure of wild type GnT-IVa was super-imposed over the crystal structure of its homolog NagH (PDB ID: 2W1U)[30] which was determined at 2.0 Å resolution with good stereochemical quality. This model includes only one Ramachandran outlier (Trp935 in chain D). The binding mode of the GlcNAc in GnT-IVa was deduced from the binding position of the GlcNAc disaccharide coordinates in the NagH crystal structure. All four glycans were docked into the GnT-IVa lectin domain by grafting β1-2GlcNAc of the α1-3 arm over the GlcNAc binding coordinates in NagH. Similarly, for three of the structures β1-4GlcNAc of the α1-3 arm was grafted over GlcNAc of NagH for preparation of an alternate binding mode. Since #103 lacks β1-4GlcNAc, for it only the β1-2GlcNAc binding mode was studied.

All seven lectin–glycan complexes were solvated in an octahedral TIP3P water box extending 10 Å from the protein surface. A total of 6 Na$^+$ ions were added to neutralize the system. A ten-step equilibration protocol published elsewhere[59] was used to equilibrate the complexes. Finally, a 500 ns MD simulation of each complex was performed at NPT, using the MD settings: temperature at 300 K, temperature scaling by Langevin dynamics (collision frequency = 2), pressure relaxation every 1.2 ps, SHAKE constraints, nonbonded interaction cutoff of 9 Å, and 2 fs integration time step. The protein was treated using the Amber ff14SB forcefield[60], whereas glycans were modelled using the Glycam06 (revision j1) forcefield[61]. Complexes were subjected to MD simulation using *cuda* version *pmemd* in Amber20[62]. MD trajectories were analysed using *cpptraj*[63]. The glycosidic dihedral angles of the chitobiose core, α1-3 arm and α1-6 arm connecting to the central mannose in each lectin–glycan complex were calculated using *cpptraj*.

The binding free energy of the glycans was calculated in both the β1-2GlcNAc and β1-4GlcNAc binding modes using the MM/GBSA approach. A total of 500 frames were extracted, one per 1 ns, for each 500 ns MD simulation and used for the binding energy calculations[64]. The GB$^{HCT}$ generalized-Born model[65] was used with set *mbondi* radii. This combination of GB model and radii outperformed other GB models in a recent study on lectin–glycan complexes[64]. The salt concentration was kept at 0.15 M. The surface tension and non-polar solvation free energy correction terms were set to 0.005 kcal·mol$^{-1}$ and 0.0, respectively. The interior (protein) and exterior (implicit solvent) dielectric constants were set to 1 and 80, respectively. Other parameters were kept at their default values in Amber20[62].

### SDS-PAGE, protein staining, lectin blotting, and Western blotting.

Proteins were separated by 5–20% SDS-PAGE. Silver staining and GelCode Blue staining were performed using Silver Stain II Kit Wako (FUJIFILM) and GelCode Blue Safe Protein Stain (ThermoFisher Scientific), respectively, in accordance with the manufacturers' protocols. For lectin and Western blotting, proteins separated by SDS-PAGE were transferred to a nitrocellulose membrane, followed by blocking with TBS-T containing 1% BSA (for lectin blotting) or TBS-T containing 5% skim milk (for Western blotting). For lectin blotting, the membranes were incubated with HRP-conjugated lectin that had been diluted with 1% BSA in TBS-T. For Western blotting, the membranes were incubated with primary and HRP-conjugated secondary antibodies that had been diluted with 5% skim milk in TBS-T. Signals were detected with Western Lightning Plus-ECL (PerkinElmer) or

SuperSignal West Femto Maximum Sensitivity substrate (Thermo Fisher Scientific) using FUSION-SOLO 7 s EDGE (Vilber-Lourmat).

**Cell culture and transfection**. COS7, HEK293, and HEK293-*MGAT4A*/*MGAT4B* double-knockout (DKO) cells were cultured with Dulbecco's modified Eagle's medium supplemented with 10% fetal bovine serum under 5% $CO_2$ conditions at 37 °C. Plasmids were transfected into cells with Lipofectamine 3000 reagent (Thermo), in accordance with the manufacturer's protocol.

**Generation of MGAT4A/MGAT4B double-knockout (DKO) cells**. DKO cells were generated using CRISPR-Cas9[66,67]. Short guide RNAs (sgRNAs) targeting human *MGAT4A* and *MGAT4B* genes were inserted into the pX330-EGFP plasmid using the oligonucleotides listed in Supplementary Table 1[68]. Parental HEK293 cells were transfected with two pX330-EGFP plasmids harboring different gRNAs targeting the human *MGAT4A* gene. Two days after transfection, cells highly expressing EGFP were collected using a FACS Melody cell sorter (BD Biosciences). To obtain DKO cells, a second round of cell sorting was conducted 2 days after the transfection of two pX330-EGFP plasmids harboring different gRNAs targeting the human *MGAT4B* gene. After the selection, a single cell clone (#12) was obtained by limiting dilution. Genotyping of DKO cells was performed by PCR using the specific primer set listed in Supplementary Table 1.

**Statistics and reproducibility**. Statistical analysis was performed using GraphPad Prism 8 software (GraphPad Software, Inc., San Diego, CA, USA). The experimental numbers (n) in this study mean biological replicates. All the blot/gel experiments were independently repeated three times and reproducibility was confirmed.

**Reporting summary**. Further information on research design is available in the Nature Research Reporting Summary linked to this article.

## Data availability

Atomic coordinates and structure factors for mouse GnT-IVa lectin domain D445A mutant have been deposited in the Protein Data Bank with accession number 7VMT. The source data behind the graphs are provided as Supplementary Data 1, and the source images for the blots/gels data are provided as Supplementary Fig. 11. Requests for plasmids and other reagents used in this study should be sent to Professor Yasuhiko Kizuka (kizuka@gifu-u.ac.jp).

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

## Acknowledgements

We are grateful to Dr. Yusuke Yamada at Photon Factory for collecting diffraction data sets and structure determination, and Ms. Keiko Hiemori at the National Institute of Advanced Industrial Science and Technology for frontal affinity chromatography analysis. We thank Prof. Seizo Koshiba, Dr. Jin Inoue and Mrs. Rika Sugai (Tohoku Medical Megabank Organization) and Tohoku University Technical Support Center for their great supports for NMR measurements at Tohoku Medical Megabank Organization. Computational modeling reported in this publication was supported by an Institutional Development Award (IDeA) from the National Institute of General Medical Sciences of the US National Institutes of Health under award number P20GM130460. We also thank Dr. Yu Kitago, Dr. Junichi Takagi, at Osaka University for consulting experiments and Dr. Sho Yamasaki at Osaka University for valuable discussions. Finally, we thank Edanz (https://jp.edanz.com/ac) for editing the English text of a draft of this manuscript. This study was supported in part by Grants for Scientific Research (C) (17K07303 and 20K06575 to M.N.) and Scientific Research (B) (20H03207 to Y.K.) from JSPS, CREST (18070267 to Y.K.) from JST, and funds from the Takeda Science Foundation to Y.K. and from the Tokyo Biochemical Research Foundation to Y.K.

## Author contributions

M.N. designed and organized the project, expressed and purified wild-type or mutated GnT-IVa lectin domain, performed the crystallographic experiments, and wrote the manuscript. T.H. established DKO cells and performed cellular experiments. H.T. performed frontal affinity chromatography analysis. S.K.M. performed molecular dynamics simulations. N.M. performed solution NMR analysis. N.O. measured the enzymatic activity, and performed cellular experiments. Y.T. measured the enzymatic activity. Y.Y. performed solution NMR analysis. R.J.D. performed molecular dynamics simulations. T.S. performed the crystallographic experiments. Y.K. designed and organized the project, measured the enzymatic activity, and wrote the manuscript. All authors commented on the manuscript and approved the submission.

## Competing interests

The authors declare no competing interests.
