## [Peer Review File · Communications Biology]

Reviewers' comments:

Reviewer #1 (Remarks to the Author):

The manuscript by Nagae and colleagues describes the characterization and crystallization of the MGAT4A lectin domain as a regulatory subunit for catalysis. MGAT4A is a potential target for the treatment of diabetes and has intriguing fundamental function in the glycosylation of various proteins and thereby this research contributes to basic as well as applied research. With this work the authors open up a path to many others researchers potentially following their insights.

- to better understand the role of MGAT4A in human health and disease, please describe known SNPs and their contribution - if data is available.

- MGAT4A is associated with diabetic phenotypes. Are there other examples for direct involvement of this enzyme? Does this still render the protein an interesting drug target for the treatment of disease?

- page 4: 4 mM Tris-HCl buffer, is this the correct value? This number seems rather low.

- same page: TPO SPIN should read TOP Spin.

- the NMR titration series, analysed by 1D 1H NMR does not generate much insight into the interaction. Changes in the NMR spectra are rather subtle, if at all. The fact that the protein can be expressed in E.coli and that it has a suitable low MW indicates that 15N labeling would be possible. Can these experiments (chemical shift perturbations) be done? Alternatively, would STD NMR lead to any meaningful saturation transfer, taking into account the low MW of the protein? Overall, the 1H data can be removed because they do not add to the overall insights and only do barely support the notion „showed greater shifts upon addition“ in the main text. If the authors insist on using these data, please show dose response curves supporting the thermodynamic constants derived from frontal affinity chromatography. And provide the protein and ligand concentration data in figure 2.

Reviewer #2 (Remarks to the Author):

The novel research in this manuscript is the discovery of a lectin domain or carbohydrate binding module in an enzyme, GnT-IVa, also designated as MGAT4. The C-terminal lectin is found to bind the product of the reaction, with GlcNAc beta1-4 linked on the alpha 1-3 Man of the central trimannose structure of N-glycans, and the lectin domain is shown to be important for the enzymatic activity, probably product release or partially modified glycoprotein, using a mutant (D445) that diminishes GlcNAc binding. The main issue is that the analyses do not follow up on the findings, for example that the binding, of GlcNAc beta1-2 linked to Man, to the lectin domain is causing greater chemical shifts than GlcNAc, but GlcNAc beta1-4 linked to Man was not used as a ligand in these experiments. GlcNAc beta1-2 linked to Man, the substrate of GnT-IVa, is not showing up as a binder in the frontal affinity chromatography and perhaps the whole enzyme should be tried here. The structure of the glycan product has to be confirmed to contain a GlcNAc beta 1-4 linkage.

Despite the realization of a crystal structure of the (mutant) lectin domain, no docking was

performed and therefore no clear view of the specific interactions of the lectin domain with the product of the GnT-IVa enzyme can be formed. The table with crystal data and structure analyses should give more focus more on the model quality than on the quantification of the outliers. The comparisons with the other proteins (NagH and IFT25/IFT27) do not go into great enough depth and remain descriptive by number pushing and not and how the glycan binding could be related to enzymatic activity and mechanisms. For example, superposition of IFT25 on NagH learns that disaccharide binding to IFT25 interferes with the long helix of the IFT27 GTPase. It would be nice to have the N-glycan symbols for beta 1-2 and beta 1-4 linkages presented in a consistent manner, preferably reflecting the same as in Figure 2C of the binding of a set of 157 glycans/polysaccharides.

Reviewer #3 (Remarks to the Author):

Review of Manuscript COMMSBIO-21-3286 by Nagae et al.

This manuscript describes the structure-function characterization of a physiologically important N-acetylglucosaminyltransferase, of which little structural and mechanistic information exists to date. This work is important and reveals interesting findings: in particular, a previously unknown lectin domain that regulates enzyme activity. The investigation is well executed and described, and spans from structural characterization by X-ray crystallography, glycan interaction studies by NMR spectroscopy, FAC and SEC, and kinetic and bioinformatic characterization of the wild-type enzyme and a single-site variant, to cellular studies.

I have a few suggestions how the manuscript can be further improved:

- Title: I am not sure if adding “a regulatory subunit for catalysis” grammatically refers to the enzyme? Maybe consider “Discovery of a lectin domain with regulatory function in” Or “Discovery of a lectin domain that regulates enzyme activity in ...”.

- The abstract and introduction are well written. What I would add is information about the size or mass of the enzyme already in the last paragraph of the introduction, to give the readers a better perspective.

- Methods: OBS! If it was the structure of the D445G variant that was solved, this should be mentioned here and in Table 1 ("native" is correct, but misleading).

Here and elsewhere in the manuscript, the language editing has gone a bit far and also changed scientific terms. For example, it should be “structure factors” (not: structural factors) in line 137, “flash-cooling” or the like in lines 127-129, Western blotting (capital W) in line 184, and “peaks” in line 242 (not: peak spectra). It would be good if the authors carefully reviewed their manuscript in this respect. I would probably also write “buffer containing 100 mM potassium iodide” on line 128/129 (I assume this was meant?) rather than 100 mM potassium iodide-containing buffer (which essentially means that we do not know how much KI is in the buffer).

Additionally, I noted that some abbreviations were not introduced, especially “CBB”, but also “Bis-Tris” (I assume Bis(2-hydroxyethyl)amino-tris(hydroxymethyl)methane?), etc.

Another small language error is in line 139, where it should be structures that are depicted using

PyMOL, or figures that are prepared with/using this program (not: figures ... depicted).

- Table 1: Generally, the information looks good, however, it should be clear if the wild-type or mutant was investigated (the PDB info suggests the latter). I am also missing the CC1/2 values and the PDB code (which is given elsewhere, but would be good to add here, too). I would further like to know how many molecules there are in the asymmetric unit, and how many protein atoms, water molecules and other atoms; as well as average B-factors and Wilson B-factor. The latter are given in the PDB report and are relatively high, with the average B-factor even being 5% higher than the Wilson B-factor. The structure was still on hold, therefore I could not check it, but I assume that it can be further improved based on the difference between average B and Wilson-B, and the large number of clashes (or are the clashes just due to hydrogen atoms being added? Why was this done?). OBS! In the PDB records, resolution limits are given instead of $1/\sigma(I)$. Table 1, Spelling: it is "figure of merit" (one r), and I assume it is one derivative (delete final e).

- Results: The results are well-described (in fact better than in the figure legends). What would be good to add is a more detailed analysis of the structure in the binding site, showing electron density for this part and checking if there may be water molecules bound, where the sugar is expected to bind. It is possible that water only bind to the wild-type structure, which would be interesting to characterize as well.

o Language, suggestion: lines 254-255: "preference for N-glycan products rather than N-glycan substrates or products that are further elongated". Line 332 (Discussion): rephrase "is supposed to have" – what is meant here? Adding a reference, may also help to understand the meaning. Line 368: we "discovered" is better than newly found; or "we discovered that the C-terminus of GnT-IVa exhibits/contains a lectin domain, which can regulate the activity of the enzyme."

- Figures: Generally, the figures are good, but I lack a picture showing the electron density, e.g., in the lectin binding site. Moreover, the legends can be and should be significantly improved. Figures 2 and 5 are the most in need of improvement, for several reasons. Fig. 2: Panel B is very small; maybe better to show an overview of the spectra above and then close-up views for different ratios below? The legend should explain what 1:0 etc stands for, and what the broken lines indicate. This is much better explained in the main text. Finally, the placement of B) and C) is very confusing, making it non-intuitive to locate the right information in the legend (swap?). Also for panel A, it was not clear what the different peaks refer to. Fig. 3: How do NagH and IFT25 come into the picture, and what are these proteins? Maybe add references also here? Again, the information given in the main text is much clearer, but the figures should also be understandable alone. Fig. 5: This figure should highlight the conclusions or hypothesis the authors arrived at, but is unclear without explanations in the legend (and for this part, also the main text is not entirely clear).

- References: The manuscript is well referenced, however, the formatting can be further improved (e.g. with respect to decapitalization, special characters, italics, etc.)

All the best,
Ute Kregel

Reviewer #1

The manuscript by Nagae and colleagues describes the characterization and crystallization of the MGAT4A lectin domain as a regulatory subunit for catalysis. MGAT4A is a potential target for the treatment of diabetes and has intriguing fundamental function in the glycosylation of various proteins and thereby this research contributes to basic as well as applied research. With this work the authors open up a path to many others researchers potentially following their insights.

1. to better understand the role of MGAT4A in human health and disease, please describe known SNPs and their contribution - if data is available.

Response

Thank you very much for your suggestion. We looked into NCBI ClinVar database (<https://www.ncbi.nlm.nih.gov/clinvar/>) and found only one coding SNP in *MGAT4A* gene (T236A). Although the clinical relevance of this variation is not clear at this moment, we mutated the corresponding residue in mouse GnT-IVa (T227A) and expressed this mutant in GnT-IVa,b-double KO cells. The *in vitro* enzyme assay clearly showed that this mutation resulted in a reduction in activity (Newly added Supplementary Figure 8B). Lectin blot with DSA also showed that this mutant produced slightly fewer glycans in cells than WT (Supplementary Figure 8A). We described these results in the Discussion as follows, "Previous studies revealed that GnT-IVa is involved in diabetes (6,22), suggesting that clarifying the regulation mechanisms of GnT-IVa activity could lead to development of a new strategy for the treatment of diabetes. Furthermore, one coding SNP (T236A) of human GnT-IVa (*MGAT4A*) was also reported in the ClinVar database (54). We found that mutation of the corresponding residue in mouse GnT-IVa (T227A) resulted in a decrease in activity in cells and *in vitro* enzyme assays (Supplementary Figure 8). Although the relevance of this SNP in specific disease is unclear at present, the reduction in activity by this coding SNP could possibly be involved in development or exacerbation of diseases." (page 11, line 14-20).

2. MGAT4A is associated with diabetic phenotypes. Are there other examples for direct involvement of this enzyme? Does this still render the protein an interesting drug target for the treatment of disease?

Response

Regarding diabetes, in addition to the diabetic phenotypes in KO mice, a previous study also showed that *MGAT4A* mRNA is reduced in pancreatic beta cells from human diabetes patients (Ohtsubo et al., *Nat. Med.*, 2011). Considering that transgenic overexpression of human *MGAT4A* in mice prevented the diabetic phenotypes induced by high fat diet (Ohtsubo et al., *Nat. Med.*, 2011), activation or upregulation of *MGAT4A* could be one of the reasonable therapeutic strategies for the treatment of diabetes. To describe this point, we added the following sentence in introduction, “Furthermore, the mRNA levels of human *MGAT4A* were also shown to be reduced in pancreatic beta cells from diabetes patients (22).” (page 3, line 4-5).

As we described in the Introduction, *MGAT4A* is also aberrantly expressed in some cancer cells, and was suggested to promote cancer cell invasion and metastasis (Nishio et al., *Oncol. Rep.*, 2017; Niimi et al., *Br. J. Cancer*, 2012; Fan et al., *Glycoconj. J.*, 2012). As we responded above, we also found a coding SNP in the ClinVar database which reduced the enzymatic activity. Future research could unveil the physiological and pathological meanings of this variation.

Therefore, available evidence suggests GnT-IVa as an interesting drug target worth studying further.

3. - page 4: 4 mM Tris-HCl buffer, is this the correct value? This number seems rather low.

Response

Thank you for pointing it out. As suggested by reviewers 1 and 2, we performed NMR experiments again with ¹⁵N-labeling and changed the figure and Methods to describe the new work. This time, the protein was dissolved in 20 mM sodium phosphate buffer, pH 6.0, as written in the Methods (page 5, line 1-3).

4. - same page: TPO SPIN should read TOP Spin.

Response

We described it correctly in the new NMR methods section. Thank you very much.

5. - the NMR titration series, analysed by 1D ^1H NMR does not generate much insight into the interaction. Changes in the NMR spectra are rather subtle, if at all. The fact that the protein can be expressed in E.coli and that it has a suitable low MW indicates that ^{15}N labeling would be possible. Can these experiments (chemical shift perturbations) be done? Alternatively, would STD NMR lead to any meaningful saturation transfer, taking into account the low MW of the protein? Overall, the ^1H data can be removed because they do not add to the overall insights and only do barely support the notion „showed greater shifts upon addition“ in the main text. If the authors insist on using these data, please show dose response curves supporting the thermodynamic constants derived from frontal affinity chromatography. And provide the protein and ligand concentration data in figure 2.

Response

As suggested, we have performed 2D NMR titration experiments with ^{15}N -labeling of the lectin domain, and the results are shown in the new Fig. 2B-D. We clearly observed the chemical shift changes in the lectin domain for both sugar ligands (GlcNAc and GlcNAc β 1-2Man), and the larger chemical shift changes and the smaller dissociation constant (3.2×10^{-4} M) were observed for GlcNAc β 1-2Man (Fig. 2B-D) relative to the GlcNAc monosaccharide (3.1×10^{-3} M). Accordingly, we have described this in the text as follows, “To confirm the direct sugar binding of GnT-IVa lectin domain, we next performed NMR titration experiments by collecting 2D ^1H - ^{15}N HSQC spectra of uniformly ^{15}N -labeled GnT-IVa lectin domain with different protein-to-ligand ratios. A limited set of protein NH signals showed apparent chemical shift change by the addition of GlcNAc (Figure 2B), reflecting the specific sugar binding to the lectin domain without global conformational change. The binding process is in the fast exchange regime in terms of chemical shift and the dissociation constant was calculated as 3.1×10^{-3} M using the NH peak A (^1H 9.84 ppm, ^{15}N 126.7 ppm) (Figure 2B and 2D). The peak A was the one that was shifted the most by the addition of the disaccharide, GlcNAc β 1-2Man, in a dose-dependent manner (Figure 2C), and the dissociation constant was calculated as 3.2×10^{-4} M using peak A (Figure 2C and 2D). This clearly indicates that the GnT-IVa lectin domain directly binds to GlcNAc-containing glycans and β 1-2 linked Man significantly contributes to its affinity.” (page 8, line 20-30).

Reviewer #2

The novel research in this manuscript is the discovery of a lectin domain or carbohydrate binding module in an enzyme, GnT-IVa, also designated as MGAT4. The C-terminal lectin is found to bind the product of the reaction, with GlcNAc beta1-4 linked on the alpha 1-3 Man of the central trimannose structure of N-glycans, and the lectin domain is shown to be important for the enzymatic activity, probably product release or partially modified glycoprotein, using a mutant (D445) that diminishes GlcNAc binding.

1. The main issue is that the analyses do not follow up on the findings, for example that the binding, of GlcNAc beta1-2 linked to Man, to the lectin domain is causing greater chemical shifts than GlcNAc, but GlcNAc beta1-4 linked to Man was not used as a ligand in these experiments. GlcNAc beta1-2 linked to Man, the substrate of GnT-IVa, is not showing up as a binder in the frontal affinity chromatography and perhaps the whole enzyme should be tried here. The structure of the glycan product has to be confirmed to contain a GlcNAc beta 1-4 linkage.

Response

Thank you very much for your reasonable comments. To obtain more insights into how GnT-IVa lectin domain recognizes glycan ligands, we newly carried out NMR titration experiments with ¹⁵N-labeling and molecular dynamics (MD) simulations of the interactions of glycan ligands to the lectin domain. Although we cannot afford and test GlcNAcβ1-4Man for NMR, our new 2D NMR results clearly showed the larger chemical shift and stronger interaction with the lectin domain for GlcNAcβ1-2Man than GlcNAc monosaccharide (Fig. 2B-D). To see how the interaction of product glycans, but not of a substrate glycan, can be detected in FAC, we performed MD simulations of the lectin domain interacting with the acceptor glycan (#103, non-binder in FAC) and the product glycans (#105, #107, and #108, all binders in FAC). Calculations of the binding free energy suggested that the β1,2-GlcNAc moiety in the non-binder can interact with the lectin domain, but its affinity is weaker than the binders (Fig. 4C). More importantly, computations of the binders showed that the β1,2-GlcNAc residue (Fig. 4C, blue) gave stronger interaction with the lectin domain than the β1,4-GlcNAc did (Fig. 4C, orange). This implies that β1,2-GlcNAc is the primary binding residue in *N*-glycans for the interactions with the GnT-IVa lectin domain and that this interaction is enhanced by the presence of β1,4-GlcNAc in the *N*-glycan. We have described the results of NMR and MD on pages 8-10.

2. Despite the realization of a crystal structure of the (mutant) lectin domain, no docking was performed and therefore no clear view of the specific interactions of the lectin domain with the product of the GnT-IVa enzyme can be formed. The table with crystal data and structure analyses should give more focus more on the model quality than on the quantification of the outliers.

Response

As we responded above, we prepared computational models of the lectin domain interacting with both binder and non-binder glycans and performed MD simulations. The results showed that the product glycans have higher affinity to the lectin domain than the acceptor glycan and that the presence of β 1,4-GlcNAc enhanced the interactions of the lectin domain with β 1,2-GlcNAc. Accordingly, we modified the text as follows, “*Molecular dynamics simulation suggests the contribution of product β 1-4GlcNAc to the tight interaction with GnT-IVa lectin domain.* To understand the structural basis of glycan recognition by the GnT-IVa lectin domain, we performed molecular dynamics (MD) simulations and analyzed the potential for binding of a non-binding biantennary glycan (#103 in Supplementary Figure 3, acceptor substrate) and three other glycans (#105, #107 and #108 in Supplementary Figure 3, product glycans) which showed strong binding in FAC. To prepare for the calculations, we positioned each glycan to have its GlcNAc overlapped with the GlcNAc from the NagH X-ray crystal structure. Glycan #103 was able to be superpositioned well over the GlcNAc of the template, without any steric clashes with the protein atoms (Supplementary Figure 5) but it drifted away from the binding site and became unbound during the subsequent MD simulations. By contrast, the other three glycans #105, #107 and #108 did show binding to the GnT-IVa lectin domain via their α 1-3 arm in the MD simulations (Figure 4A, B and Supplementary Figure 6). In bisected glycan #108 the α 1-6 arm is unlikely to bind to the lectin domain, as the α 1-6 arm back-flips towards the chitobiose core and therefore is not exposed enough for recognition by proteins (53). Assuming glycan binding to a particular lectin domain has a common molecular recognition mechanism for all the binding glycans, the likelihood of glycan binding to the lectin domain via the α 1-6 arm is low and hence we omitted it. It is evident from MD that the α 1-3 arm of #105, #107 and #108 can bind to the GnT-IVa lectin domain in a manner in which β 1-2GlcNAc or β 1-4GlcNAc occupies the primary GlcNAc binding site. The molecular mechanics/generalized-Born surface area (MM/GBSA) binding energies show stronger binding affinity when β 1-2GlcNAc occupies the primary binding site compared to β 1-4GlcNAc (Figure 4C; Supplementary Table 2). This suggests that the presence of β 1-4GlcNAc in the α 1-3 arm is needed for glycan binding, even if it may not be interacting directly with the lectin domain. It is likely that β 1-4GlcNAc plays a key role in stabilizing the conformation of the overall glycan structure and contributes favorably to the entropic contribution. This explanation is supported by the glycan conformations in the protein bound states

that show that the chitobiose core is more flexible and adopts two major conformations in the case of the non-binding glycan #103 compared to the case of the binders (Supplementary Figure 7). This shows that the presence of β 1-4GlcNAc in the α 1-3 arm stabilizes the ligand conformation.” (page 9, last line-page10, line 26).

As for Table 1, we have added “Favored” and “Allowed” regions in Ramachandran plot and clashscore of Molprobit output, according to the reviewer’s suggestion. Thank you very much.

3. The comparisons with the other proteins (NagH and IFT25/IFT27) do not go into great enough depth and remain descriptive by number pushing and not and how the glycan binding could be related to enzymatic activity and mechanisms. For example, superposition of IFT25 on NagH learns that disaccharide binding to IFT25 interferes with the long helix of the IFT27 GTPase.

Response

Thank you for important comment. We made a new figure to show the structural comparison in putative sugar binding sites among three proteins, GnT-IVa, NagH, and IFT25/27 (Figure 3E) and described the structural details of each structure. Accordingly, we modified the text as follows, “A DALI search revealed that the 3D structure of the GnT-IVa lectin domain shows high structural similarity to those of NagH [PDB code: 2W1U, (38)] and IFT25/27 complex [PDB code: 2YC4, (52)] (Figure 3D). The structural superpositions with NagH (Z-score = 11.9) showed that the RMSD value of the corresponding 126 C α atoms was 2.6 Å, whereas the superposition of IFT25, a component of the IFT25/27 complex (Z-score = 11.9), indicated that the RMSD of the corresponding 115 C α atoms was 2.6 Å. As predicted from sequence analysis, the structural comparison with NagH GlcNAc-containing disaccharide (GlcNAc β 1-3GalNAc) complex demonstrated that amino acid residues that directly interact with the GlcNAc at the non-reducing end in NagH are completely conserved in the GnT-IVa lectin domain (Figure 3E, left and middle panels). Four residues (Y819, W836, D877, and W935) are completely conserved in GnT-IVa (Y394, W410, D445, and W513). Thus, this indicates that the GnT-IVa lectin domain also binds to GlcNAc moieties in sugar ligands at the same site. However, several loop regions of GnT-IVa are slightly apart from those of NagH (Supplementary Figure 4A). The position of D445A is slightly buried and there is no water molecule corresponding to the hydroxyl group of GlcNAc around D445A (Supplementary Figure 4B). Note, though, that these loops contact neighboring molecules in the crystal packing, and such artificial interaction may affect the local structure (Supplementary Figure 4C). In contrast, the overall structure of IFT25 is similar to that of the GnT-IVa lectin domain, but the corresponding region

of the putative sugar binding site in IFT25 shows marked contrast with the other two proteins (Figure 3E, right panel). The aspartate and three aromatic residues are not conserved. Instead, a long α -helix of IFT27 fully occupies the sugar binding site. This clearly explains the functional difference of GnT-IVa and IFT25.” (page 9, line 17-line 2 from the bottom)

4. It would be nice to have the N-glycan symbols for beta 1-2 and beta 1-4 linkages presented in a consistent manner, preferably reflecting the same as in Figure 2C of the binding of a set of 157 glycans/polysaccharides.

Response

According to your suggestion, we changed the N-glycan symbols in Fig. 1A, 1F, 5C, 6, S1A, S8, and S9.

Reviewer #3

This manuscript describes the structure-function characterization of a physiologically important N-acetylglucosaminyltransferase, of which little structural and mechanistic information exists to date. This work is important and reveals interesting findings: in particular, a previously unknown lectin domain that regulates enzyme activity. The investigation is well executed and described, and spans from structural characterization by X-ray crystallography, glycan interaction studies by NMR spectroscopy, FAC and SEC, and kinetic and bioinformatic characterization of the wild-type enzyme and a single-site variant, to cellular studies.

I have a few suggestions how the manuscript can be further improved:

1. Title: I am not sure if adding “a regulatory subunit for catalysis” grammatically refers to the enzyme? Maybe consider “Discovery of a lectin domain with regulatory function in” Or “Discovery of a lectin domain that regulates enzyme activity in ...”.

Response

Thank you for your suggestion. We have changed the title as suggested, “**Discovery of a lectin domain that regulates enzyme activity in *N*-acetylglucosaminyltransferase-IVa (MGAT4A)**”.

2. The abstract and introduction are well written. What I would add is information about the size or mass of the enzyme already in the last paragraph of the introduction, to give the readers a better perspective.

Response

Thank you very much for your suggestion. We have added the following sentence in Introduction “GnT-IVa (Q812G0 in UniProt) is composed of 526 amino acids with the predicted size of 60.6 kDa” (page 3, line 13-14).

3. Methods: OBS! If it was the structure of the D445G variant that was solved, this should be mentioned here and in Table 1 (“native” is correct, but misleading).

Response

Thank you for pointing it out. We have clarified “D445A mutant” in the Methods (page 4, line 17 and 21) and “D445A mutant (PDB ID: 7VMT)” in Table 1.

4. Here and elsewhere in the manuscript, the language editing has gone a bit far and also changed scientific terms. For example, it should be “structure factors” (not: structural factors) in line 137, “flash-cooling” or the like in lines 127-129, Western blotting (capital W) in line 184, and “peaks” in line 242 (not: peak spectra). It would be good if the authors carefully reviewed their manuscript in this respect. I would probably also write “buffer containing 100 mM potassium iodide” on line 128/129 (I assume this was meant?) rather than 100 mM potassium iodide-containing buffer (which essentially means that we do not know how much KI is in the buffer).

Response

Thank you for pointing them out. We have changed “structural factors” to “structure factors”, “directly frozen with liquid nitrogen” to “directly flash-cooled in liquid nitrogen”, “rapid freezing” to “flash cooling”, and “western blotting” to “Western blotting”. We also changed “100 mM potassium iodide-containing buffer” to “0.1 M Bis-tris (pH 6.5), 0.2 M magnesium chloride, 25% (w/v) polyethylene glycol 3,350, and 0.1 M potassium iodide” (page 4, lines 21-22).

5. Additionally, I noted that some abbreviations were not introduced, especially “CBB”, but also “Bis-Tris” (I assume Bis(2-hydroxyethyl)amino-tris(hydroxymethyl)methane?), etc.

Response

We spelled out these abbreviations in the text. Thank you very much.

6. Another small language error is in line 139, where it should be structures that are depicted using PyMOL, or figures that are prepared with/using this program (not: figures ... depicted).

Response

We have changed “figures” to “structures” “by” to “using”. Thank you.

7. Table 1: Generally, the information looks good, however, it should be clear if the wild-type or mutant was investigated (the PDB info suggests the latter). I am also missing the CC1/2 values and the PDB code (which is given elsewhere, but would be good to add here, too). I would further like to know how many molecules there are in the asymmetric unit, and how many protein atoms, water molecules and other atoms; as well as average B-factors and Wilson B-factor. The latter are given in the PDB report and are relatively high, with the average B-factor even being 5% higher than the Wilson B-factor. The structure was still on hold, therefore I could not check it, but I assume that it can be further improved based on the difference between average B and Wilson-B, and the large number of clashes (or are the clashes just due to hydrogen atoms being added? Why was this done?). OBS! In the PDB records, resolution limits are given instead of $1/\sigma(I)$. Table 1, Spelling: it is “figure of merit” (one r), and I assume it is one derivative (delete final e).

Response

As we responded above, we have mentioned in Table 1 that D445A mutant was crystalized. In Table 1, we also added the PDB code, CC1/2 value, the number of molecules in the asymmetric unit, the number of atoms (protein, water and others), average B factor (protein, water and others), and Wilson B-factor. As the reviewer

pointed out, the average B factor of our model (42.0 \AA^2) is slightly higher than that of Wilson B-factor (36.8 \AA^2). This is probably because our model includes several highly mobile loop-short helix regions (D460-G476) and one mobile molecule (please see Figure R1). In these regions, the peak height of the electron density is relatively low, but observed. So, we carefully put the polypeptide chains in these regions. The stereochemistry of these regions is located within the acceptable region. Thus, we thought this relatively high B-factor is not a critical issue for publication.

Figure R1: Visualization of B-factor in the asymmetric unit. Six molecules, small molecules and solvent are colored by B-factor using the default settings in PyMOL (spectrum b).

Furthermore, we have corrected spelling of “Figure of Merit”, and deleted the final “e” from “derivatives”. Thank you very much.

8. Results: The results are well-described (in fact better than in the figure legends). What would be good to add is a more detailed analysis of the structure in the binding site, showing electron density for this part and checking if there may be water molecules bound, where the sugar is expected to bind. It is possible that water only bind to the wild-type structure, which would be interesting to characterize as well.

Response

Thank you for your good suggestion. As mentioned in the response to reviewer 2, we have added a figure showing electron density map around the sugar binding site (Supplementary Figure 4B). We also added the structural details of the putative sugar binding site as well as water molecule around D445A. We modified the structural description as follows, “A DALI search revealed that the 3D structure of the GnT-IVa lectin domain shows high structural similarity to those of NagH [PDB code: 2W1U, (38)] and IFT25/27 complex [PDB code: 2YC4, (52)] (Figure 3D). The structural superpositions with NagH (Z-score = 11.9) showed that the RMSD value of the corresponding 126 C α atoms was 2.6 Å, whereas the superposition of IFT25, a component of the IFT25/27 complex (Z-score = 11.9), indicated that the RMSD of the corresponding 115 C α atoms was 2.6 Å. As predicted from sequence analysis, the structural comparison with NagH GlcNAc-containing disaccharide (GlcNAc β 1-3GalNAc) complex demonstrated that amino acid residues that directly interact with the GlcNAc at the non-reducing end in NagH are completely conserved in the GnT-IVa lectin domain (Figure 3E, left and middle panels). Four residues (Y819, W836, D877, and W935) are completely conserved in GnT-IVa (Y394, W410, D445, and W513). Thus, this indicates that the GnT-IVa lectin domain also binds to GlcNAc moieties in sugar ligands at the same site. However, several loop regions of GnT-IVa are slightly apart from those of NagH (Supplementary Figure 4A). The position of D445A is slightly buried and there is no water molecule corresponding to the hydroxyl group of GlcNAc around D445A (Supplementary Figure 4B). Note, though, that these loops contact neighboring molecules in the crystal packing, and such artificial interaction may affect the local structure (Supplementary Figure 4C). In contrast, the overall structure of IFT25 is similar to that of the GnT-IVa lectin domain, but the corresponding region of the putative sugar binding site in IFT25 shows marked contrast with the other two proteins (Figure 3E, right panel). The aspartate and three aromatic residues are not conserved. Instead, a long α -helix

of IFT27 fully occupies the sugar binding site. This clearly explains the functional difference of GnT-IVa and IFT25.” (page 9, line 17-line 2 from the bottom).

To obtain structural insights into how the lectin domain recognizes glycan ligands, we further performed MD simulations using binders and non-binders. As shown in Figure 4, we showed that the beta1-2GlcNAc primarily binds to D445 and that this interaction is enhanced by the presence of beta1-4GlcNAc.

9. o Language, suggestion: lines 254-255: “preference for N-glycan products rather than N-glycan substrates or products that are further elongated”. Line 332 (Discussion): rephrase “is supposed to have” – what is meant here? Adding a reference, may also help to understand the meaning. Line 368: we “discovered” is better than newly found; or “we discovered that the C-terminus of GnT-IVa exhibits/contains a lectin domain, which can regulate the activity of the enzyme.”

Response

Thank you very much for these suggestions. We changed “preference~” to “preference for short *N*-glycan products rather than *N*-glycan substrates or products that are further elongated”. We also changed “GnT-III is supposed to have~” to “GnT-III is predicted to have a GT-A fold with DXD motif”, and a reference (Nagae et al., *Int. J. Mol. Sci.*, 2020, 21, 437) was added to this sentence. We also changed “newly found” to “discovered” as suggested.

10. Figures: Generally, the figures are good, but I lack a picture showing the electron density, e.g., in the lectin binding site. Moreover, the legends can be and should be significantly improved. Figures 2 and 5 are the most in need of improvement, for several reasons. Fig. 2: Panel B is very small; maybe better to show an overview of the spectra above and then close-up views for different ratios below? The legend should explain what 1:0 etc stands for, and what the broken lines indicate. This is much better explained in the main text. Finally, the placement of B) and C) is very confusing, making it non-intuitive to locate the right information in the legend (swap?). Also for panel A, it was not clear what the different peaks refer to. Fig. 3: How do NagH and IFT25 come into the picture, and what are these proteins? Maybe add references also here? Again, the information given in the main text is much clearer, but the figures should also be understandable alone. Fig. 5: This figure should highlight the conclusions or hypothesis

the authors arrived at, but is unclear without explanations in the legend (and for this part, also the main text is not entirely clear).

Response

Thank you very much for your reasonable comments. We have added the electron density map around the ligand binding site (Supplementary Figure 4B), and mentioned this figure in the text as follows, “The position of D445A is slightly buried and there is no water molecule corresponding to the hydroxyl group of GlcNAc around D445A (Supplementary Figure 4B).” (page 9, lines 28-30).

Fig. 2: To address the concerns raised by reviewer 1 and 2, we have performed NMR experiments again with ¹⁵N-labeling and changed Figure 2B to a new version (Fig. 2B-D). Now we see clear chemical shifts of a peak (Peak A in Fig. 2B and 2C). We also described the ligand-to-protein ratio in the Figure legends. The placements of the figures (Fig. 2B-E) were also changed. In Fig. 2A, to explain more about the differently eluted peaks, we have added the following sentence in the legend, “The same gel filtration analysis was performed in the presence of 2 mM Glc (green) or GlcNAc (red).”

Fig. 3: To explain the reason why we showed NagH and IFT25, we changed the figure legend of Fig. 3D as follows: “(D) Structural neighbors of GnT-IVa lectin domain defined by DALI. Overall structures of NagH (PDB code: 2W1U, (38), left panel) and IFT25 (PDB code: 2YC4, (52), right panel), which are structurally similar proteins to the GnT-IVa lectin domain. These two proteins are viewed from the same angle as Fig. 3C. The carbohydrate and calcium ion are shown in stick and sphere models, respectively.”

Fig. 6 (previous Fig. 5): We added the following explanation in Fig. 6 legend, “Our data showed that the lectin domain specifically binds *N*-glycan products and is required for efficient GnT-IVa reaction. We propose that the lectin domain is involved in either (i) the prompt product release in the catalytic reaction cycle or (ii) the efficient modification of glycoproteins in which β1-4GlcNAc already exists.”

11. - References: The manuscript is well referenced, however, the formatting can be further improved (e.g. with respect to decapitalization, special characters, italics, etc.)

Response

We have modified reference formatting. Thank you very much.

Note that we also made a few other minor edits to improve the English, which also are shown

in red for the ease of review.

Reviewers' comments:

Reviewer #1 (Remarks to the Author):

All concerns have been addressed by the authors.

Reviewer #3 (Remarks to the Author):

It is nice to see this manuscript further improved. I have a few final comments that the authors may wish to take into account when preparing their final manuscript.

Mainly this concerns the refinement of the crystal structure. The authors point out that the structure contains several highly mobile loops, but this will also affect the Wilson B-factor (in fact more so, since not all residues are modeled). It is therefore highly likely that the structure can be further refined, even though the R-factors indicate that the structure is overall correct already now, and that the main conclusions in the manuscript will not be affected. I cannot comment on the quality of the X-ray structure in more detail without access to the structural data. However, certainly more details in the Methods section would be appreciated, e.g. if restraints (or constraints) were used between the 6 molecules in the asymmetric unit, and which protocol was used for refining the B-factors. Potentially, TLS refinement could help to further improve the structure. – Regarding Table 1, I appreciate that the authors added the Wilson B-factor and CC1/2 values, but wonder if there was a typo for the high-resolution value of the iodide derivative, as a CC1/2 < 0.1 does not contribute meaningful information. The figure of merit, with 0.24 (please delete last 2 digits), is also extremely low, but it seems that the authors nevertheless managed to solve the crystal structure, which is similar to other structures in the PDB. – I recommend to replace the word “redundancy” with “multiplicity” in this Table. The clash score could be removed to save space.

Also in need of correction, on page 7: “Hirata et al., submitted elsewhere” is not helpful, and not correct in the context of “as described previously”.

Additional small comments:

- Last paragraph of introduction: please replace “the predicted size of 60.6 kDa” with “a predicted mass of 60.6 kDa”.

- Page 3, Methods, Plasmid construction: “The plasmid for soluble and full-length D445A mutants and full-length...” – a) What exactly is meant with soluble D445A mutant – which construct? Can the authors refer to one of the constructs displayed in a figure? b) I would probably add “of GnT-IVa”. – c) Also: “mutant” usually refers to a substitution in a gene, whereas one usually refers to “protein variants”. Consider changing throughout the manuscript.

- Page 4: I suggest to call Figure 3A after “Coomassie Brilliant Blue staining” (the abbreviation CBB is not really necessary, neither in the text nor the figure).

- Next paragraph: consider specifying further which construct was crystallized, e.g. “All crystallization trials of the putative lectin domain (variant D445A, residues 382-526)”.

- Page 6, Computational modeling: It would be good to comment on the quality and resolution of the homolog NagH (PDB ID: 2W1U), which is very good.
- Page 7, last paragraph: also give sequence identity for the central part of GnT-IVa (17%), rather than the non-defined term "similarity". Both 14% and 17% are very low.
- Page 9, first paragraph: I suggest to call Figure 2E after "GlcNACbeta1-2Man".
- Page 13, Author contributions: add "which was approved by all authors" (and MD simulations should be plural).
- References: decapitalize all references and double check for missing italics etc. e.g. in ref. 38.
- Supplementary Table 2: units are missing.
- Supplementary Figure 4A: I appreciate the new figure with electron density. Even better would be a white background and a choice of orientation that matches one of the other panels, or a superimposition to see where the water molecule would be expected. In the legend, it should be "contoured at 1.5 sigma", not s.
- Supplementary Figure 5: Consider revising figure caption to "Initial binding mode..." to highlight that the glycan drifted away, since the figure itself suggests otherwise.
- Supplementary Figure 6: Consider revising figure caption to "Snapshots off binding modes..."

Reviewer #1

Comment

All concerns have been addressed by the authors

Response

Thank you very much for your constructive comments. We hope that the revised paper is now acceptable.

Reviewer #3

Comment 1

It is nice to see this manuscript further improved. I have a few final comments that the authors may wish to take into account when preparing their final manuscript.

Response

Thank you very much your careful and extensive reviews. We revised the paper according to your comments, and hope that the revised version is now acceptable.

Comment 2

Mainly this concerns the refinement of the crystal structure. The authors point out that the structure contains several highly mobile loops, but this will also affect the Wilson B-factor (in fact more so, since not all residues are modeled). It is therefore highly likely that the structure can be further refined, even though the R-factors indicate that the structure is overall correct already now, and that the main conclusions in the manuscript will not be affected. I cannot comment on the quality of the X-ray structure in more detail without access to the structural data. However, certainly more details in the Methods section would be appreciated, e.g. if restraints (or constraints) were used between the 6 molecules in the asymmetric unit, and which protocol was used for refining the B-factors. Potentially, TLS refinement could help to further improve the structure. – Regarding Table 1, I appreciate that the authors added the Wilson B-factor and CC1/2 values, but wonder if there was a typo for the high-resolution value of the iodide derivative, as a $CC1/2 < 0.1$ does not contribute meaningful information. The figure of merit, with 0.24 (please delete last 2 digits), is also extremely low, but it seems that the authors

nevertheless managed to solve the crystal structure, which is similar to other structures in the PDB. – I recommend to replace the word “redundancy” with “multiplicity” in this Table. The clash score could be removed to save space.

Response

Thank you very much for your kind suggestions. As suggested, we tried to refine the final model with TLS parameter option in phenix.refine. Unfortunately, we could not dramatically improve the quality of our model. We have added the details of refinement in the Method section (page 4) and corrected Table 1. In addition, we carefully checked Table 1 and found a lot of mistakes in the statistics of iodide derivatives as pointed out. We apologized for these mistakes and really appreciated your careful inspection.

Comment 3

Also in need of correction, on page 7: “Hirata et al., submitted elsewhere” is not helpful, and not correct in the context of “as described previously”.

Response

Thank you for pointing it out. We have corrected the text and added the title of the paper as follows, “DKO cells were generated as described in our paper (Hirata et al., “*Shedding of N-acetylglucosaminyltransferase-V is regulated by maturity of cellular N-glycan*”) using CRISPR-Cas9” (page 7)

Additional small comments:

Comment 4

- Last paragraph of introduction: please replace “the predicted size of 60.6 kDa” with “a predicted mass of 60.6 kDa”.

Response

We have corrected as suggested. Thank you.

Comment 5

- Page 3, Methods, Plasmid construction: “The plasmid for soluble and full-length D445A mutants and full-length...” – a) What exactly is meant with soluble D445A mutant – which construct? Can the authors refer to one of the constructs displayed in a figure? b) I would probably add “of GnT-IVa”. – c) Also: “mutant” usually refers to a substitution in a gene, whereas one usually refers to “protein variants”. Consider changing throughout the manuscript.

Response

a) We are sorry for our poor explanation. As you assumed, the soluble construct is pcDNA-IH/GnT-IVa 60-526 which appeared in Fig. 5A. We have clearly described this construct name in the method. Now the sentence is, “The plasmids for soluble (pcDNA-IH/GnT-IVa 60–526) and full-length (pcDNA6/mycHisA/GnT-IVa) D445A mutants and full-length T227A mutant of GnT-IVa”. b) We have also added “of GnT-IVa”, as suggested. c) We think that “mutant” can also be used for referring to a protein derived from a mutated gene. Therefore, we would like to use mutant in this manuscript.

Comment 6

- Page 4: I suggest to call Figure 3A after “Coomassie Brilliant Blue staining” (the abbreviation CBB is not really necessary, neither in the text nor the figure).

Response

We have added “Figure 3A” as suggested.

Comment 7

- Next paragraph: consider specifying further which construct was crystallized, e.g. “All crystallization trials of the putative lectin domain (variant D445A, residues 382-526)”.

Response

We have added the residue numbers as follows, “All crystallization trials of the lectin domain D445A mutant (residues 382-526)”. Thank you very much.

Comment 8

- Page 6, Computational modeling: It would be good to comment on the quality and resolution of the homolog NagH (PDB ID: 2W1U), which is very good.

Response

We added the following sentence regarding the resolution and quality of NagH in the Method section, “which was determined at 2.0 Å resolution with good stereochemical quality. This model includes only one Ramachandran outlier (Trp935 in chain D).”.

Comment 9

- Page 7, last paragraph: also give sequence identity for the central part of GnT-IVa (17%), rather than the non-defined term “similarity”. Both 14% and 17% are very low.

Response

We meant it identity and corrected “similarity” to “low identity” in the text (page 7).

Comment 10

- Page 9, first paragraph: I suggest to call Figure 2E after “GlcNACbeta1-2Man”.

Response

Thank you very much. We have added Figure 2E, as suggested.

Comment 11

- Page 13, Author contributions: add “which was approved by all authors” (and MD simulations should be plural).

Response

We have corrected it. Thank you.

Comment 12

- References: decapitalize all references and double check for missing italics etc. e.g. in ref. 38.

Response

We checked the reference list again and corrected several references.

Comment 13

- Supplementary Table 2: units are missing.

Response

We have added the missing units. Thank you.

Comment 14

- Supplementary Figure 4A: I appreciate the new figure with electron density. Even better would be a white background and a choice of orientation that matches one of the other panels, or a superimposition to see where the water molecule would be expected. In the legend, it should be “contoured at 1.5 sigma”, not s.

Response

Thank you very much for your suggestions. We redrew the electron density map around D445A in Suppl. Fig. 4B so that the revised figure was depicted from the same view angle as Suppl. Fig. 4A. We also corrected the Figure legend.

Comment 15

- Supplementary Figure 5: Consider revising figure caption to “Initial binding mode...” to highlight that the glycan drifted away, since the figure itself suggests otherwise.

Response

We agree with you and have added “Initial” in the legend.

Comment 16

- Supplementary Figure 6: Consider revising figure caption to “Snapshots off binding modes...”.

Response

We have added "Snapshots of" in the legend. Thank you very much.